# Could Student Selection Be the Missing Piece for Efficient Distillation?

## Abstract

Selecting the optimal student architecture remains an overlooked challenge in knowledge distillation (KD). Current approaches typically rely on model size constraints or random selection, ignoring how student architecture and inductive biases impact distillation effectiveness. We formulate this as an unsupervised model selection problem, where the goal is to select the best student for a given teacher without requiring ground-truth labels or expensive training cycles. We propose a transferability metric based on the Neural Tangent Kernel (NTK) that quantifies function space alignment between teacher and student models. Specifically, our cross-model NTK measures the directional similarity between teacher and student gradient vectors on unlabeled data, capturing how effectively the student can mimic the teacher's function through gradient-based optimization. Unlike existing transferability metrics that require ground-truth labels and focus on model-dataset relationships, our approach directly models the model-model relationship central to KD. To ensure practical applicability with modern networks, we implement an efficient approximation using Johnson-Lindenstrauss random projections that preserves gradient inner products without computing full NTK matrices. Experiments demonstrate that our metric is robust, and reliably predicts the post-distillation performance, outperforming existing transferability scores adapted for KD and baseline selection strategies, even in low-data scenarios. Our approach enables efficient identification of compatible student architectures before training, eliminating the need for resource-intensive trial-and-error in model compression pipelines. Our code can be found in https://anonymous.4open.science/r/C-MoNA-7CE5.

## 1 Introduction

Knowledge distillation (KD) (Hinton et al., 2015) enables transferring knowledge from a high-capacity teacher model to a smaller student model, typically by training the student to match the teacher's predicted class probabilities. While the KD procedure itself is well-established, student model selection, the task of choosing an optimal student for a given teacher model, remains largely ad hoc, often based solely on size constraints or random choice. This approach is suboptimal since the student's architecture and inductive biases significantly impact its ability to learn from a given teacher (Mirzadeh et al., 2020; Cho & Hariharan, 2019). Different architectures encode information differently, causing substantial variation in the student's capacity to distill the teacher's knowledge. This motivates the need for a principled method to predict, before training, how well a candidate student will perform under distillation from a specific teacher, especially in the case of downstream task adaptation using KD-based fine-tuning.

Recent theoretical advances in distillation scaling laws (Busbridge et al., 2025) reveal that student performance depends not only on the student's own capacity, but crucially on the capacity gap between teacher and student - a relationship entirely absent in supervised learning. This demonstrates that rankings derived from model size constraints cannot reliably predict distillation performance, since the teacher-dependent optimization landscape fundamentally alters the performance dynamics. The selection of optimal student architecture becomes even more critical in paradigms like mutual learning (Zhang et al., 2018; Yao & Sun, 2020) and online collaborative learning (Guo et al., 2020), where distillation occurs bidirectionally and teacher-student mismatches can compound across multiple learning iterations.

Table 1: Comparison of transferability estimation approaches. Traditional methods focus on model-data alignment and require labels, while our model-model alignment metric directly measures teacher-student gradient alignment in a label-free manner. "✓" indicates full support, "∼✓" partial support, and "×" no support.

|                            | Label-Free | KD-Compatible | Low-Data Robustness | Model–Data |
| -------------------------- | ---------- | ------------- | ------------------- | ---------- |
| **Transferability Estimation** | ∼✓      | ×             | ×                   | ✓          |
| **Student Model Selection**    | ✓       | ✓             | ✓                   | ×          |

While traditional transferability estimation approaches (Ding et al., 2022; Nguyen et al., 2020; Wang et al., 2023; Gholami et al., 2023; You et al., 2021; Khoba et al., 2025) provide promising solutions for model selection given particular datasets, they focus on model-data relationships and ignore the crucial teacher-student (model-model) compatibility needed for KD-based fine-tuning, potentially preventing full utilization of teacher knowledge either through logit-based or feature-based distillation. Unlike supervised fine-tuning where optimization dynamics are determined by hard labels, knowledge distillation creates an optimization landscape fundamentally dependent on the teacher's dynamics and soft targets, necessitating assessment of student-teacher learning alignment rather than conventional model-data relationships. In KD, we operate in a label-free scenario where the only supervision comes from the teacher's outputs on unlabeled data, creating a gap in the literature for unsupervised transferability estimation specifically tailored to student-teacher compatibility.

To address this gap, we propose an unsupervised transferability metric based on Neural Tangent Kernel (NTK) (Jacot et al., 2018) theory. We introduce a **cross-model NTK** that quantifies teacher-student function space alignment by directly comparing the teacher's and student's gradient vectors for the same inputs. This measures directional similarity between the models: if the student's parameter updates point in similar directions to the teacher's gradients, the student is inherently well-aligned to mimic the teacher's function. Our cross-model NTK approach differs fundamentally from existing methods that compare individual NTKs (Harutyunyan et al., 2023). Rather than evaluating models independently, our approach directly measures the compatibility between student and teacher gradient structures, bypassing the need for pseudo-label generation or explicit label access. This provides a critical advantage in privacy-constrained scenarios that other estimation metrics fail to offer.

To ensure practical applicability with modern networks, we implement an efficient algorithm using Johnson-Lindenstrauss random projections (Rahimi & Recht, 2007) that preserves gradient inner products while avoiding computationally-expensive full Jacobian calculations, enabling scalable evaluation of multiple candidate students. Our contributions are four-fold:

- We formalize the problem of unsupervised student model selection for knowledge distillation, establishing the theoretical foundation for why model-model compatibility cannot be predicted from model-data relationships and demonstrating the limitations of supervised-learning-derived rankings for distillation scenarios.

- We propose C-MoNA, a transferability estimation metric based on cross-model NTK alignment. This is the first approach to employ neural tangent kernels across two models to quantify their distillation compatibility. Our scalable implementation using random projections makes the method feasible for modern large-scale networks while maintaining complete privacy preservation.

- Through extensive experiments across image classification and object detection tasks using multiple distillation strategies, we demonstrate that our method consistently outperforms existing transferability estimation metrics adapted for KD and baseline selection strategies.

- We demonstrate the robustness of our method in privacy-constrained, low-data scenarios where the number of samples is significantly smaller than the number of classes, showing strong performance with minimal degradation while existing transferability metrics fail to produce reliable results.

## 2 PROBLEM MOTIVATION

Like existing transferability estimation methods, our work addresses the practical need to predict model performance without expensive training cycles, reducing computational costs and time-to-deployment. The ability to identify compatible teacher-student pairs (both pre-trained on respective datasets) before fine-tuning becomes critical when one must select students from large candidate pools under computational budgets and privacy constraints that prohibit extensive empirical evaluation.

### 2.1 THEORETICAL FOUNDATION: WHY MODEL-DATA METRICS FAIL FOR KD

**The Model-Data Paradigm of Existing Metrics.** Existing transferability metrics like LogME (You et al., 2021), LEEP (Nguyen et al., 2020), and PACTran (Ding et al., 2022) follow a common paradigm where transferability is measured as $f(\phi(X), Y)$—assessing how well a pre-trained model's feature representations $\phi(X)$ align with target labels $Y$ through mutual information, correlation, or Bayesian evidence. This model-data alignment assumption works for traditional transfer learning but fails to capture the model-model compatibility essential for knowledge distillation.

**The Model-Model Paradigm Required for Knowledge Distillation.** Knowledge distillation operates under a fundamentally different paradigm where success depends on how effectively a student model can mimic a teacher's function, not on model-data alignment. The distillation scaling law (Busbridge et al., 2025) formalizes this distinction:

$$L_S(N_S, D_S, L_T) = \underbrace{L_T}_{\text{Teacher cross-entropy}} + \underbrace{\frac{1}{L_T^{c_0}}\left(1 + \frac{L_T}{\tilde{L}_S d_1}\right)^{-c_1 f_1}\left(\frac{A}{N_S^{\alpha'}} + \frac{B}{D_S^{\beta'}}\right)^{\gamma'}}_{\text{Student ability to mimic teacher}} \tag{1}$$

where $\mathcal{L}_S$ is student distillation loss, $N_S$ and $D_S$ are student model size and data size, $L_T$ is teacher loss, $\tilde{L}_S$ is student supervised loss, and $c_0, c_1, f_1, d_1, A', B', \alpha', \beta', \gamma'$ are empirically fitted scaling constants. The equation decomposes into three components: teacher performance baseline ($L_T$), teacher-dependent distillation improvement (middle term), and student compute scaling (final term). This reveals three components absent from model-data metrics: (1) **Teacher Performance Dependency** ($L_T$) where student performance is directly conditioned on teacher capability, (2) **Capacity Gap Interactions** ($L_T/\tilde{L}_S$) where the teacher-student performance ratio creates multiplicative effects that alter student rankings, and (3) **Non-linear Coupling** where the capacity gap term modulates student scaling through complex non-linear functions, meaning student capability alone cannot predict distillation success.

**Mathematical Proof of Ranking Divergence.** Consider two students $S_1$ and $S_2$ with supervised performances $L_{S_1}^e < L_{S_2}^e$ (i.e., $S_1$ outranks $S_2$ under supervision). Their distillation performances are $\mathcal{L}_{S_i} = L_T + g\left(\frac{L_T}{L_{S_i}^e}\right) \cdot h(N_{S_i}, D_{S_i})$ for $i \in \{1, 2\}$, where $g(\cdot)$ denotes capacity gap function . Ranking reversal occurs when $\mathcal{L}_{S_1} > \mathcal{L}_{S_2}$ despite $L_{S_1}^e < L_{S_2}^e$, requiring:

$$g\left(\frac{L_T}{\tilde{L}_{S_1}}\right) \cdot h(N_{S_1}, D_{S_1}) > g\left(\frac{L_T}{\tilde{L}_{S_2}}\right) \cdot h(N_{S_2}, D_{S_2}) \tag{2}$$

Since $\tilde{L}_{S_1} < \tilde{L}_{S_2}$ implies $\frac{L_T}{\tilde{L}_{S_1}} > \frac{L_T}{\tilde{L}_{S_2}}$, ranking reversals occur when $g(\cdot)$ coupling with student scaling terms creates unfavorable interactions for stronger supervised students. Please refer to the section A.3 for detailed analysis.

**Empirical Evidence from Supervision Complexity Theory.** The supervision complexity framework (Harutyunyan et al., 2023) provides additional theoretical evidence for why model-data alignment fails to predict distillation success. Their analysis shows that distillation effectiveness depends on the alignment between teacher-provided supervision and the student's Neural Tangent Kernel:

$$\text{Distillation Success} \propto \text{Alignment}(\text{Teacher Supervision}, \Theta_{\text{student}}(x, x')) \tag{3}$$

This alignment is fundamentally a model-model property that cannot be inferred from how either model individually relates to ground-truth labels.

**The Inadequacy of Size-Based Heuristics and Implications for Student Selection.** Current practice relies on model size as a compatibility proxy, but scaling laws reveal that capacity gaps depend on loss values $L_T/L_S^e$, not parameter counts. Models with identical architectures can exhibit vastly different $L_S^e$ values due to pretraining data quality, training procedures, and architectural inductive biases, making size-based selection systematically unreliable. Effective student selection therefore requires metrics that (a) capture model-model relationships rather than model-data alignments, (b) account for teacher-specific compatibility rather than general transferability, and (c) base decisions on learned representations rather than model size. These requirements motivate our cross-model NTK approach, which directly measures gradient-space alignment between teacher and student models—precisely the model-model relationship that determines distillation success.

## 3 PROPOSED APPROACH

### 3.1 PROBLEM FORMULATION

Given a large pre-trained teacher model $f_t(x; \theta_t)$ trained on a target distillation dataset $D = \{x_n\}_{n=1}^N$ ($N$: number of samples), the objective is to select the ideal student model/rank student models from a set of candidate student models $\{\Phi_m\}_{m=1}^M$ based on their distillation ability. The setup considers that no pre-training data of the teacher or the student model is known or provided. We assume a fixed teacher model fine-tuned on the target dataset, with both teacher and student pre-trained on their respective datasets, positioning our work within the knowledge distillation fine-tuning paradigm. We focus on efficient fine-tuning regimes (50-100 epochs) for practical deployment, contrasting with extensive protocols using 200-700 epochs (Beyer et al., 2022). While our formulation selects optimal students for a given teacher, the methodology can extend to joint teacher-student optimization with increased computational complexity.

### 3.2 PRELIMINARIES

The Neural Tangent Kernel (NTK) (Jacot et al., 2018) establishes a connection between neural networks and kernel methods, showing that infinitely wide networks trained with gradient descent behave like kernel machines with a specific kernel determined by the architecture at initialization. This framework enables predicting the final function learned by the network and its generalization properties by analyzing the NTK properties, explaining why very large networks can be trained successfully despite highly non-convex optimization landscapes—in the infinite-width limit, the dynamics become effectively linear in function space.

Formally, for a deep neural network $f(x; \theta)$ with parameters $\theta$, the NTK is defined as:

$$\Theta(x, x') = \nabla_\theta f(x; \theta)^\top \cdot \nabla_\theta f(x'; \theta) \tag{4}$$

where $\nabla_\theta f(x; \theta)$ denotes the gradient of the network's output with respect to its parameters, evaluated at input $x$. Applying the chain rule layer-wise, we get the very early empirical NTK defined as:

$$\Theta(x, x') = \sum_{l=1}^L \left( \frac{\partial f(x; \theta)}{\partial f^{(l)}(x; \theta)} \left( f^{(l-1)}(x; \theta) \right)^\top \right)^\top \cdot \left( \frac{\partial f(x'; \theta)}{\partial f^{(l)}(x'; \theta)} \left( f^{(l-1)}(x'; \theta) \right)^\top \right) \tag{5}$$

where $L$ is the total number of layers used in the computation. Crucially, the NTK's ability to predict network behavior based on initialization properties aligns directly with our predictive setting for student selection. Since the NTK encapsulates the inductive biases inherent in network architectures and captures per-example similarities without requiring training, it provides a natural foundation for assessing model compatibility in an unsupervised manner before any knowledge distillation occurs. This initialization-based predictive capability makes NTK theory particularly well-suited for our goal of identifying compatible teacher-student pairs prior to expensive training cycles.

### 3.3 C-MONA: CROSS-MODEL NTK ALIGNMENT

Given a fixed neural network for teachers $f_t(x; \theta_t)$, we consider the parameters of the teacher network $\theta_t$ as the frozen reference and for a given candidate student model $f_s(x; \theta_{s_0})$, we consider $\theta_{s_0}$ as the parameters of the student models at random initialization. NTK comparison between two models

examines and contrasts the fixed (or nearly fixed) gradient-propagation geometry of two architectures, revealing how well they would learn in the "lazy" or linearized regime.

The naive way to do this is to compare the individual NTKs $f_t$ and $f_s$ obtained by using Eq. (5) using Centered Kernel Alignment (CKA) as defined in Eq. (6):

$$\text{CKA}(\Theta_{t,t}, \Theta_{s,s}) = \frac{\langle \Theta_{t,t}, \Theta_{s,s} \rangle_F}{\|\Theta_{t,t}\|_F \cdot \|\Theta_{s,s}\|_F} \tag{6}$$

But, this leads to suboptimal performance for the goal of picking the best student model. The downside of naively considering CKA as a student model selection metric is that it acts as a generic similarity measure of the matrices representing the initial dynamics. This is a step removed from comparing the raw activations or the raw gradient vectors directly, therefore offering suboptimal model selection performance. Since the teacher and student models are pre-trained on substantially different datasets, their resulting weights differ significantly. Consequently, directly comparing their kernel matrices introduces a considerable amount of random noise, diminishing the reliability of such comparisons. Table 9 empirically shows the suboptimal performance of CKA when compared to the proposed metric. We argue that our proposed metric Cross-Model NTK Alignment (C-MoNA) is substantially a better way to map the model-model relationship.

Mathematically, cross-model NTK is defined as:

$$\Theta_{t,s}(x, x') = \nabla_{\theta_t} f_t(x; \theta_t)^\top \cdot \nabla_{\theta_s} f_s(x'; \theta_s) \tag{7}$$

Applying chain-rule layer-wise, the very early empirical cross-model NTK is defined as:

$$\Theta_{t,s}(x, x') = \sum_{l=1}^{L} \left( \frac{\partial f_t(x; \theta_t)}{\partial f_t^{(l)}(x; \theta_t)} \left( f_t^{(l-1)}(x; \theta_t) \right)^\top \right)^\top \cdot \left( \frac{\partial f_s(x'; \theta_s)}{\partial f_s^{(l)}(x'; \theta_s)} \left( f_s^{(l-1)}(x'; \theta_s) \right)^\top \right) \tag{8}$$

The cross-model NTK matrix is a $N \times N$ gram matrix full of cross-gradients, which allows us to inspect the structure (spectra, eigenvectors) to see which modes of the candidate student model are most aligned with the reference teacher model. Intuitively, it provides us a cross-sensitivity, *i.e,* entry $(i, j)$ in $\Theta_{t,s}(x, x')$ tells us how a tiny step on $t@x_i$ moves $s$'s output $@x_j$. It can be thought of as a full cross-covariance of the two Jacobian maps, not per-input equality test.

Since we need a scalar metric to estimate the transferability score, we have to reduce the large $N \times N$ matrix to a scalar in an interpretable manner. The choice of matrix reduction strategy is theoretically motivated by the nature of knowledge distillation itself. Knowledge distillation requires *global function compatibility* rather than point-wise correspondence—students must learn the teacher's overall response pattern across the input space, which is captured by cross-sample gradient correlations in the off-diagonal terms of the NTK matrix. From NTK theory, the complete kernel matrix encodes the model's inductive bias and learning dynamics, where trace-based norms measure only self-alignment energy while the Frobenius norm captures total cross-alignment energy between teacher and student learning manifolds. Supporting empirical results are provided in the appendix.

Therefore, we do a global alignment based on Frobenius-norm. The idea being very simple and straight-forward, we treat $\Theta_{t,s}(x, x')$ as a big matrix and compare its Frobenius norm to those of the self-NTKs: $\|\Theta_{t,s}\|_F$ vs. $\sqrt{\|\Theta_{t,t}\|_F \cdot \|\Theta_{s,s}\|_F}$. Therefore, our transferability estimation metric C-MoNA is defined as:

$$\alpha = \frac{\|\Theta_{t,s}\|_F}{\sqrt{\|\Theta_{t,t}\|_F \cdot \|\Theta_{s,s}\|_F}} \in [0, 1] \tag{9}$$

$\alpha \approx 1$ implies that the two learning geometries are almost indistinguishable in aggregate, while $\alpha \approx 0$ means that they are nearly orthogonal. As seen in Eq. (9), C-MoNA adopts a formulation similar to Eq. (6), but applies it directly on the Jacobians to provide a more granular view of gradient alignment. Taking the Frobenius norm of $\Theta_{t,s}$ focuses on point-wise gradient similarity across the teacher and student models rather than the "self-influence" structure within each model that their self-NTKs $\Theta_{t,t}$ and $\Theta_{s,s}$ capture.

Therefore, $\Theta_{t,s}$ is very different from $\text{CKA}(\Theta_{t,t}, \Theta_{s,s})$ as it directly compares the gradient directions in the teacher's parameter space to the gradient directions in the student's parameter space making it

directly compatible for student model selection for KD as it measures if the student "wants to change" in a similar direction as the teacher for a given input. Concisely, the cross-Jacobian matrix in Eq. (9) captures the overall strength of alignment or the shared energy between all pairs of teacher gradients and student gradients across all data points. Even though we use the basic NTK formulation to prove our hypothesis, advanced variants (Zhou & Zhu, 2024; Xu et al., 2021) that improve kernel regression fidelity could further enhance transferability estimation accuracy.

**Scalable Implementation via Johnson-Lindenstrauss Lemma.** Modern teacher models like ViT (Dosovitskiy et al., 2020), ConvNext-Large (Liu et al., 2022), and Wide-ResNet (Zagoruyko & Komodakis, 2016) have hundreds of millions of parameters, making full Jacobian matrix calculation for Eq. (9) computationally prohibitive. We adopt random projection approximation (Engel et al., 2023; Park et al., 2023) using the Fast Johnson-Lindenstrauss transform (FJLT) (Rahimi & Recht, 2007) to efficiently reduce dimensionality while preserving gradient inner products essential for NTK computation.

Formally, we project gradients $\nabla_{\theta_t} f_t(x; \theta_t)$ and $\nabla_{\theta_s} f_s(x'; \theta_s)$ from high-dimensional spaces $\mathbb{R}^d$ to low-dimensional space $\mathbb{R}^k$ with $k \ll d$. The Johnson-Lindenstrauss projection uses structured random projection matrices $R : \mathbb{R}^d \mapsto \mathbb{R}^k$ with $k = O\left(\frac{\log n}{\varepsilon^2}\right)$, where $n$ is the number of data samples, such that for any vectors $u, v$:

$$(1 - \varepsilon)\|u - v\|^2 \leq \|Ru - Rv\|^2 \leq (1 + \varepsilon)\|u - v\|^2 \tag{10}$$

with high probability. This distance preservation directly translates to preserving gradient inner products, the fundamental building blocks of NTK computation.

**Handling Heterogeneous Architectures:** For teacher and student models with different parameter dimensions, we employ Rademacher projection matrices $R_t$ and $R_s$ drawn from the same underlying distribution using a fixed random seed, ensuring consistent statistical properties for meaningful comparison in the projected space $\mathbb{R}^k$.

Using Johnson-Lindenstrauss projection, we modify Eq. (7) to use projected gradients:

$$\nabla_{\theta_t} f_t(x; \theta_t)^\top \nabla_{\theta_s} f_s(x'; \theta_s) \approx \left(R_t \nabla_{\theta_t} f_t(x; \theta_t)\right)^\top \left(R_s \nabla_{\theta_s} f_s(x'; \theta_s)\right) \tag{11}$$

The proposed metric inherently depends only on the the per-example similarity constituted by the geometries of the models in question making it label space agnostic and thereby an effective unsupervised and scalable transferability estimation metric for mapping model-model relationship.

## 4 EXPERIMENTS

### 4.1 EXPERIMENTAL SETTINGS

**Image Classification:** The study evaluates 4 teachers and 12 students spanning diverse architectures and pretraining corpora; teachers are pretrained on different datasets and finetuned on the target dataset. To be precise, the teacher models comprise of CLIP-ViTB/32 (Radford et al., 2021; Ilharco et al., 2021) image encoder, MaxViT-S (Tu et al., 2022), Wide-ResNet101 (Zagoruyko & Komodakis, 2016), and ConvNext-Large (Liu et al., 2022) spanning CNN-only, CNN-Transformer hybrid, and Transformer-only architecture types with different pre-training data and learning objectives. As for the student models, for the ease of experimentation, all the models were pretrained on the ImageNet dataset. The student models include ResNet34 (He et al., 2016), ResNet18 (He et al., 2016), ShuffleNetV2 (Ma et al., 2018), EfficientNet-B0 (Tan & Le, 2019), MobileNet-V2 (Sandler et al., 2018), DeiT-T/16 (Touvron et al., 2021), DeiT-T/16-distilled (Touvron et al., 2021), DeiT-S/16 (Touvron et al., 2021), DeiT-S/16-distilled (Touvron et al., 2021), SwinT-Tiny (Liu et al., 2021), GoogleNet (Szegedy et al., 2015), and MnasNet (Tan et al., 2019), spanning different model capacity, architecture families and therefore the inductive biases. We conduct experiments on three image classification benchmarking datasets, like the CIFAR100 (Krizhevsky et al., 2009), Caltech101 (Fei-Fei et al., 2004) and the SUN397 (Xiao et al., 2010) datasets. Ground-truth accuracies are obtained using logit-based KD with KL divergence as the objective function.

**Object Detection.** For the object detection task, we primarily vary the pre-training datasets for the teachers and student models. Inspired by ETran (Gholami et al., 2023), VOC2012 (Everingham et al., 2011) is split into a 12-class source cluster and an 8-class target cluster. From the source cluster,

3 classes are randomly selected and repeated 12 times to form 12 source datasets; a YOLOv11s (Khanam & Hussain, 2024) student is trained on each, yielding 12 diverse pretrained students. Teachers are YOLOv11l and YOLOv11m pretrained on COCO (Lin et al., 2014), HomeObjects3k (Jocher & Rizwan, 2025), and AfricanWildlife (Jocher & Rizwan, 2025), producing 4 diverse teacher models. Ground-truth accuracies are obtained via SimKD's (Chen et al., 2022) approach using feature-based L2 distillation. These settings span diverse architectures and pretraining data, rigorously validating the metric across datasets, tasks, and distillation strategies. Additional details can be found in the appendix.

**Training Setup Differences.** For classification distillation, the final layer was trainable and C-MoNA applied to the entire model worked effectively. For object detection using SimKD, the teacher's detection head was frozen and reused in the student model, with only backbone features being trained. In this case, applying C-MoNA to the entire model hurt performance, but applying it only to trainable layers was effective.This suggests that C-MoNA's flexibility to target specific layers makes it effective across different training configurations where only parts of the model are being updated, which other estimation metrics lack as they mostly measure feature clustering in the penultimate layer.

## 4.2 Evaluation Criteria

Given that we have the ground-truth ranking scores of all the student candidate models in the model hub with respect to each teacher model, we use the Kendall's tau (Kendall, 1938), $\tau$ as our main criteria to evaluate C-MoNA, similar to the previous works in the transferability estimation community(Shao et al., 2022; Ding et al., 2022; You et al., 2021; Khoba et al., 2025). Kendall's tau(Kendall, 1938) is a rank-based correlation metric that quantifies the degree of agreement between two ordered lists. It is computed by taking the difference between the number of concordant and discordant pairs and normalizing by the total number of possible pairs, $M(M-1)/2$. Formally, the Kendall's $\tau$ is:

$$\tau = \frac{2}{M(M-1)} \sum_{i=1}^{M} \sum_{j=i+1}^{M} \text{sgn}(G_i - G_j) \cdot \text{sgn}(T_i - T_j) \tag{12}$$

where $G$ and $T$ denote the two ranked sequences ($G$ being the ground truth rankings obtained via KD-based fine-tuning and $T$ being the estimated rankings based on different metrics in question), and $\text{sgn}(\cdot)$the sign function indicating the direction of difference. Following (Gholami et al., 2023), we employ the weighted version of the Kendall's tau(Shieh, 1998), $\tau_w$ which assigns greater importance to disagreements among higher-ranked items, thereby emphasizing alignment at the top of the ranked list, which is crucial in practice such as in model selection for recommendation systems.

## 4.3 Experimental Results and Benchmarks

Since most of the existing transferability metrics require the ground truth labels to compute their transferability score, we generate pseudo labels from the teacher model as ground truth labels for these methods. Our proposed metric is not dependent on the labels and therefore does not require us to store any of the teacher model's features making it unsupervised and effective when there exists privacy concerns. To the best of the author's knowledge, the only other unsupervised metric available is ETran's Energy score(Gholami et al., 2023), which is label agnostic.

**Full-data regime.** In this model selection phase, we consider that we have access to the images of the entire train-set of the distillation dataset. We benchmark C-MoNA with the other state-of-the-art transferability methods and the results are shown in Tables 2, 3, and 4. From the empirical results it is clearly evident that none of the previous methods work well across diverse teachers as they simply model the model-data relationship and ignore the model-model relationship that is key in knowledge distillation based fine-tuning. The proposed C-MoNA outperforms all the SOTA estimation methods by a significant margin across all the image classification datasets. Across all the datasets, the only other truly unsupervised metric ETran Energy (Gholami et al., 2023) shows negative correlation. This highlights the importance of considering the shared energies between teacher and student models in knowledge distillation. Relying solely on the student's energy is insufficient, as the shared energy better reflects how effectively the student model can distill the teacher's outputs.

**Low-data regime.** To showcase the reliability and robustness of our metric, we benchmark C-MoNA with other estimation metrics in a low data regime using only a handful of images ranging from

10 images to 100 images in total picked randomly from the target dataset. This regime is not very uncommon and is very much practically applicable when there are privacy concerns regarding the dataset. Most importantly, this setting does not guarantee that there is at least one image sample per class making most of the previous work highly unreliable. This setting truly reflects the effectiveness of C-MoNA over the existing ones as most of the prior methods are based on discriminant analysis. Table 6 clearly shows that the proposed C-MoNA is least sensitive to the number of samples provided and outperforms the existing works by a very large margin on varied random image subsets of 10 images, 25 images, and 50 images. Fig. 1 presents the sensitivity analysis of C-MoNA's ranking performance (Kendall's tau) across teacher models on three datasets with varying numbers of unlabeled samples. The error bars demonstrate that C-MoNA stabilizes with just 25-50 unlabeled samples, making it practical for deployment in privacy-constrained or limited-data scenarios.

Since VOC2012 dataset only has 8 target classes in our setup, we have benchmarked C-MoNA with other estimation metrics only in the low-data regime mostly on the random 100 images subset. Table 5 demonstrates C-MoNA's effectiveness on object detection tasks, which involves the added complexity of regression alongside classification. Despite requiring no supervision, C-MoNA achieves $1.3\times$ improvement over the best-performing transferability estimation metric.

Table 7 shows the actual post-distillation peformance of the C-MoNA selected student versus the best baseline methods' selected student model. It shows that C-MoNA achieves near-oracle performance while the best baseline method lags considerably behind, exhibiting significant instability across diverse teachers (evidenced by higher standard deviations). Similar results on other datasets can be found in the appendix. To further quantify the effectiveness of our metric, we check the top-5 hits across teachers and datasets for the best baseline and the C-MoNA metric. This helps in the reduction of search space where more hits implies the metric is good at avoiding worse performing models. Table 8 clearly showcases the consistent superior performance of the metric across teacher models and datasets when compared to the best baseline. C-MoNA consistently achieves $\geq 3/5$ top-ranked hits across datasets and teachers displaying reliability which the best existing method fails to achieve. Ablation studies, additional experimental results along with all the per teacher scores for all of the settings are provided in the appendix. The ground truth accuracies are provided in the appendix.

Table 2: Kendall's tau ($\tau_w$) of different transferability estimation methods on the SUN397 dataset assessing the estimation fidelity in KD based fine-tuning paradigm across diverse teacher models.

| | Wide-ResNet101 | ConvNext-Large | MaxViT-Small | OpenCLIP-ViT-B/32 | Mean±Std |
|---|---|---|---|---|---|
| NLEEP | 0.105 | -0.295 | 0.333 | 0.037 | 0.045±0.260 |
| LogME | -0.183 | 0.096 | -0.090 | -0.207 | -0.096±0.138 |
| SFDA | 0.257 | 0.163 | -0.208 | 0.346 | 0.139±0.243 |
| ETran Energy | -0.397 | -0.444 | -0.471 | -0.359 | -0.417±0.050 |
| PACTran | -0.074 | 0.219 | 0.174 | -0.044 | 0.068±0.149 |
| NCTI | -0.574 | -0.268 | -0.228 | -0.366 | -0.359±0.155 |
| SA(SFDA) | 0.265 | 0.090 | 0.202 | 0.308 | 0.216±0.095 |
| C-MoNA (ours) | **0.348** | **0.353** | **0.378** | **0.367** | **0.361±0.014** |

Table 3: Kendall's tau ($\tau_w$) of different transferability estimation methods on the CIFAR100 dataset assessing the estimation fidelity in KD based fine-tuning paradigm across diverse teacher models.

| | Wide-ResNet101 | ConvNext-Large | MaxViT-Small | OpenCLIP-ViT-B/32 | Mean±Std |
|---|---|---|---|---|---|
| NLEEP | 0.286 | 0.279 | **0.555** | -0.031 | 0.272±0.240 |
| LogME | -0.436 | -0.564 | -0.650 | -0.152 | -0.450±0.218 |
| SFDA | 0.175 | -0.165 | -0.308 | -0.078 | -0.094±0.203 |
| ETran Energy | -0.177 | -0.139 | -0.144 | -0.236 | -0.174±0.045 |
| PACTran | 0.027 | -0.186 | -0.234 | 0.105 | -0.072±0.164 |
| NCTI | -0.153 | 0.081 | -0.002 | 0.136 | 0.015±0.126 |
| SA(NLEEP) | -0.408 | -0.476 | -0.343 | -0.326 | -0.388±0.068 |
| C-MoNA (ours) | **0.536** | **0.569** | 0.456 | **0.371** | **0.483±0.089** |

## 5 RELIABILITY OF TRANSFERABILITY ESTIMATION METRICS

In the low-data regime, when the number of samples are significantly less compared to the number of classes, methods based on discriminant analysis often encounter numerical stability issues. Prior

Table 4: Kendall's tau ($\tau_w$) for the Caltech101 dataset using full trainset.

|  | MaxViT-Small | Wide-ResNet101 | ConvNext-Large | OpenCLIP-ViT-B/32 | Mean±Std |
|---|---|---|---|---|---|
| ETran Energy | -0.306 | -0.221 | -0.459 | -0.329 | -0.328±0.095 |
| SFDA | -0.081 | 0.017 | **0.338** | -0.184 | 0.022±0.209 |
| LogME | -0.204 | **0.231** | 0.281 | -0.231 | 0.019±0.248 |
| PAC | -0.065 | -0.022 | 0.047 | -0.261 | -0.075±0.127 |
| NCTI | -0.184 | -0.001 | -0.757 | -0.398 | -0.335±0.289 |
| NLEEP | 0.103 | -0.147 | -0.410 | 0.062 | -0.098±0.230 |
| SA(SFDA) | -0.160 | -0.501 | -0.520 | -0.114 | -0.323±0.196 |
| C-MoNA (ours) | **0.261** | 0.079 | 0.114 | **0.210** | **0.166±0.079** |

Table 5: Kendall's tau ($\tau_w$) for the object detection task evaluated on VOC2012 images.

|  | Yolo11l-Coco | Yolo11l-HomeObj | Yolo11m-HomeObj | Yolo11m-Wild | Mean±Std |
|---|---|---|---|---|---|
| ETran Energy | -0.200 | -0.242 | -0.558 | -0.454 | -0.363±0.148 |
| LogME | -0.278 | 0.250 | -0.218 | -0.197 | -0.11±0.210 |
| SFDA | 0.370 | **0.391** | 0.072 | 0.028 | 0.215±0.166 |
| PAC | 0.169 | 0.026 | 0.253 | **0.521** | 0.242±0.180 |
| NCTI | 0.078 | 0.209 | 0.315 | 0.290 | 0.223±0.093 |
| NLEEP | 0.380 | -0.236 | 0.210 | 0.458 | 0.203±0.269 |
| CKA | 0.037 | -0.152 | -0.428 | -0.391 | -0.233±0.189 |
| C-MoNA (ours) | **0.509** | 0.236 | **0.421** | 0.122 | **0.322±0.152** |

Table 6: Kendall's tau ($\tau_w$) for model selection with minimal data ($<< 1\%$ of dataset) averaged across 4 teachers.

| Method | SUN397 |  |  |  | CIFAR100 |  |  |  | Caltech101 |  |  |  |
|---|---|---|---|---|---|---|---|---|---|---|---|---|
|  | 10 | 25 | 50 | Avg. | 10 | 25 | 50 | Avg. | 10 | 25 | 50 | Avg. |
| ETran Energy | -0.301 | -0.294 | -0.269 | -0.288 | -0.107 | -0.168 | -0.146 | -0.140 | -0.325 | -0.309 | -0.159 | -0.264 |
| SFDA | - | - | 0.191 | - | - | 0.348 | 0.321 | - | - | 0.030 | -0.011 | - |
| LogME | -0.156 | -0.198 | -0.161 | -0.172 | -0.188 | -0.306 | -0.282 | -0.259 | -0.049 | -0.041 | -0.030 | -0.040 |
| PACTran | -0.062 | 0.051 | -0.213 | -0.075 | -0.259 | -0.144 | -0.218 | -0.207 | -0.109 | 0.013 | 0.006 | -0.030 |
| NCTI | - | - | - | - | - | -0.214 | -0.134 | - | - | -0.214 | 0.170 | - |
| **C-MoNA** | **0.059** | **0.366** | **0.290** | **0.238** | **0.414** | **0.452** | **0.465** | **0.444** | **0.159** | **0.332** | **0.313** | **0.268** |

works based on Linear Discriminant Analysis (LDA), in particular, is highly dependent on the label space, as it collapses within-class variability and focuses solely on maximizing class separation. When labeled examples are sparse, the directions learned by LDA can become noisy or even misaligned. This makes LDA sensitive not only to the quantity of available data but also to the quality of the teacher model, which provides pseudo-labels used as ground truth. If the number of labeled samples is limited and the teacher's pseudo labels are unreliable, LDA's covariance estimation becomes ill-posed, since it requires a sufficient number of samples per class to compute a stable and meaningful matrix. This explains the occurrence of NaNs in certain settings for methods based on discriminant analysis, raising concerns about the reliability of these metrics in real-world scenarios where data privacy is a critical constraint. Another example of a data-hungry metric is NLEEP (Li et al., 2021), which depends on computing class-wise statistics, particularly class-conditional covariance matrices. When the number of samples is significantly smaller than the number of classes, these covariance matrices become singular (non-invertible), making eigenvalue computation unstable or undefined. As a result, the metric could not be reliably computed in the low-data regime, and thus have been excluded from the corresponding experimental results. For methods not based on LDA, PACTran (Ding et al., 2022) includes hyperparameters—most notably $\lambda$ in the PAC-Bayes bound—that make its estimates sensitive to the specific choice of samples. While C-MoNA also benefits from diverse subsets, it remains simple and reliable in practice, avoiding invalid scores when sample diversity is limited. Accordingly, for object detection, results are reported on a randomly sampled 100-image subset; with only 50 images, most competing metrics fail to produce valid scores. On the other hand, NTK gives us a full kernel perspective that captures the rich structure in image space (smoothness, invariance,etc.) without the requirement of the label space. It does not depend on knowing the number of classes as well for the kernel computation making it a good choice for privacy-constrained

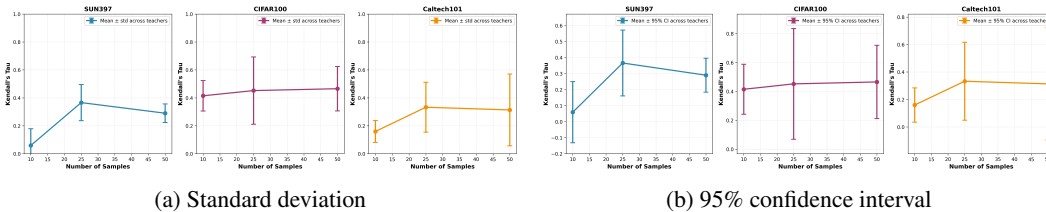

(a) Standard deviation                             (b) 95% confidence interval

Figure 1: Sensitivity analysis of Kendall's $\tau_w$ correlation as a function of sample size across three datasets (SUN397, CIFAR100, Caltech101). Error bars represent (a) standard deviation and (b) 95% confidence intervals computed across multiple teacher models. The metric stabilizes with 25–50 samples for most datasets.

Table 7: Task performance when using the selected models for the best two methods post KD fine-tuning on SUN397 dataset across different teacher models.

| Method | MaxViT-Small | WideResNet-101 | ConvNeXt-Large | OpenCLIP-ViT-B/32 | Mean±Std |
|---|---|---|---|---|---|
| SFDA | 62.39 | 62.27 | 67.98 | **67.24** | 64.97±3.06 |
| C-MoNA (ours) | 67.78 | 65.50 | 67.47 | 66.54 | 66.82±1.03 |
| Oracle | **68.30** | **66.74** | **68.47** | 67.24 | **67.69±0.83** |

Table 8: Benchmarking top-5 Hits for the best baseline method against C-MoNA across different datasets. Avg is floored to ensure strict evaluation.

| Datasets | Method | WRN101 | ConvNeXt-L | MaxViT-S | CLIP-ViT-B/32 | Avg. |
|---|---|---|---|---|---|---|
| Caltech101 | SFDA | 2/5 | 2/5 | 1/5 | 1/5 | 1/5 |
|  | C-MoNA (ours) | 3/5 | 3/5 | 3/5 | 3/5 | **3/5** |
| CIFAR100 | NLEEP | 3/5 | 3/5 | 4/5 | 2/5 | 3/5 |
|  | C-MoNA (ours) | 4/5 | 4/5 | 3/5 | 3/5 | **3/5** |
| SUN397 | SFDA | 2/5 | 2/5 | 2/5 | 2/5 | 2/5 |
|  | C-MoNA (ours) | 3/5 | 3/5 | 3/5 | 4/5 | **3/5** |

settings. Therefore, the NTK similarities reflect intrinsic data geometry that can be well aligned across models. Moreover, the NTK based alignment is more inherent to the architecture and width, than the dataset size. This makes NTK-based alignment methods like C-MoNA a reliable and robust choice for assessing model compatibility, which is essential for student model selection in knowledge distillation-based fine-tuning in privacy-constrained, low-data scenarios. As shown in Tables 3 and 6, the C-MoNA metric exhibits an average performance drop of less than 9% in the low-data regime (using significantly less than 1% of the training set) compared to the full-data setting on the CIFAR100 dataset. In contrast, several other metrics suffer substantial performance degradation, while those based on discriminant analysis often fail to produce reliable results, frequently resulting in NaNs. Remarkably, in the low-data regime, the C-MoNA metric outperforms previous state-of-the-art methods—designed to capture only the model-data relationship—even when those methods are evaluated on their full datasets, achieving gains of 1.8× improvement on SUN397, 1.6× on CIFAR100, and over 11× improvement on the Caltech101 image classification datasets with 3-5× lower variance. This highlights both the robustness of the proposed metric and the critical importance of capturing model-to-model relationships for effective knowledge distillation-based fine-tuning.

## ETHICS STATEMENT

This work fully complies with the ICLR Code of Ethics. All research activities were performed with honesty, transparency, and a commitment to reproducibility. Experimental results are truthfully presented. No personal, sensitive, or proprietary data were used, and there are no privacy or licensing concerns associated with this study. No foreseeable harm arises from the work, prior literature and funding sources are cited, and there are no conflicts of interest. The paper does not infringe on institutional, legal, or ICLR ethical standards.

## REPRODUCIBILITY STATEMENT

Significant efforts were made to ensure the reproducibility of all findings. The main body includes thorough descriptions of each model, training routine, and evaluation strategy. Additional implementation details and hyperparameters are provided in the appendix. All datasets are openly accessible. Experiment code is provided to facilitate the replication of our results.

## LLM USAGE STATEMENT

Large language models (LLMs) were used only for minor editorial tasks, such as refining grammar and improving linguistic clarity. LLMs had no role in research ideation, methodology, experiment design, implementation, analysis, or formulating scientific claims. All theoretical insights, experiments, and conclusions were created solely by the authors.

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

# A APPENDIX

## A.1 RELATED WORK

### A.1.1 KERNEL-BASED KNOWLEDGE DISTILLATION METHODS

A growing body of work has explored kernel and correlation matrix-based approaches during knowledge distillation training to improve the distillation process itself. Correlation Congruence (CC) Peng et al. (2019) transfers instance-wise correlation matrices between teacher and student, measuring pairwise similarity across mini-batches to align correlation structures. Class Attention Transfer (CAT) Guo et al. (2023) uses spatial attention maps as kernel-like structures to guide knowledge transfer from intermediate teacher layers to student layers. Similarity-Preserving Knowledge Distillation (SP) Tung & Mori (2019) employs activation-based similarity matrices to preserve pairwise instance relationships during distillation training.

Recent advances include methods that transfer full kernel matrices effectively Qian et al. (2022), local correlation consistency frameworks Li et al. (2020), and kernel-alignment based approaches Saha et al. (2022); Zhou et al. (2024). While Centered Kernel Alignment (CKA) based approaches have proven to be effective for comparing representations during distillation training, we demonstrate in Section 4.3 that direct application of CKA between individual teacher and student NTKs performs poorly as a predictive metric for student model selection. These kernel-based methods differ fundamentally from our approach in several key aspects: (1) **Temporal usage**: Prior methods use kernel/correlation matrices *during* KD training as loss functions to improve the distillation process, while we use kernel matrices *before* KD training for compatibility assessment; (2) **Captured knowledge**: Existing methods focus on feature/output similarities through activation correlations, whereas our approach measures gradient space alignment via cross-model projected gradient inner products; (3) **Computational phase**: Traditional kernel-based KD operates within the training loop, while our method enables pre-selection before expensive training begins. Our approach is orthogonal and complementary to existing kernel-based KD losses, serving a fundamentally different purpose of student selection rather than distillation enhancement.

### A.1.2 TRANSFERABILITY ESTIMATION

Numerous transferability estimation metrics have been proposed to predict downstream performance without full fine-tuning. Methods like LEEP Nguyen et al. (2020), LogME You et al. (2021), and PACTran Ding et al. (2022) require ground-truth labels to assess model-data alignment, measuring how well pre-trained features align with target decision boundaries or estimating fine-tuning risk through PAC-Bayesian bounds. While SFDA Shao et al. (2022) attempts an unsupervised approach by approximating Fisher Discriminant, it simulates class separability and fails with limited samples per class. Recent approaches like energy-based ETran Gholami et al. (2023), Neural Collapse Transferability Index (NCTI) Wang et al. (2023), OTCE Tan et al. (2021), and WDJE Zhan & Zeng (2023) broaden applicability beyond classification through energy scores, neural collapse analysis, optimal transport, and Wasserstein distances respectively, but fundamentally still assess model-dataset relationships rather than model-model compatibility essential for knowledge distillation.

As shown in Table 1, they assess model–data alignment, ranking pre-trained networks by how well their features or predictions fit the target labels. They do not consider model–model compatibility and typically degrade when label information is scarce or absent. In contrast, our work addresses student model selection for knowledge distillation, where one must choose among multiple pre-trained student architectures given a fixed fine-tuned teacher and unlabeled data. We propose a *cross-model NTK* metric that directly measures gradient-space alignment between teacher and student on the same inputs, addressing the model-model relationship central to KD. Our cross-model NTK differs fundamentally from simply comparing individual NTKs as done in Harutyunyan et al. (2023) to verify online distillation effectiveness, instead capturing how readily a student can learn this specific teacher's outputs.

## A.2 CONCLUSION AND FUTURE WORK

In this work, we first formalize the problem of unsupervised student model selection for KD, and then introduce a novel transferability estimation metric, C-MoNA, based on cross-model Neural Tangent Kernel (NTK) alignment to capture the model-to-model relationship. We demonstrate the

scalability of our approach across large-scale networks, highlighting its practical relevance. Through comprehensive experiments, our results clearly underscore the importance of modeling inter-model compatibility for effective student model selection in KD-based fine-tuning.

While the proposed NTK-based metric outperforms existing methods, there is still considerable room for improvement. Based on the assumption of linear training dynamics, linear interpolation can be used to approximate the teacher's near-pretrained weights. Leveraging the teacher model's pre-fine-tuned weights for alignment may yield a more reliable and informative estimation metric. However, since teacher and student models are often pre-trained on different datasets, their weights may lie in incompatible regions of the parameter space. Additionally, extending this approach to multi-modal LLMs presents an intriguing direction, though it may require careful handling of modality-dependent contributions that vary based on input queries. Exploring these different perspectives may open up promising directions for future research work.

### A.3 A PLAUSIBLE EXAMPLE OF RANKING DIVERGENCE

Equation 1 is an **empirically fitted scaling law** (Busbridge et al., 2025). It is not a closed-form solution but gives a *theoretically grounded* model for how teacher quality, student supervised loss, model size and data size interact and scale under distillation. We use it for conceptual illustration and support its conclusions with experiments.

In Eq. (1), the term

$$\left(1 + \frac{L_T}{\tilde{L}_S} \frac{1}{d_1}\right)^{-c_1 f_1}$$

controls the *capacity–gap penalty*: $c_1 f_1$ sets its strength and $d_1$ shifts the gap scale at which the penalty activates, while $L_T$ and $\tilde{L}_S$ determine the teacher–student mismatch.

The multiplicative factor

$$\frac{1}{L_T^{c_0}}$$

is a *teacher-amplifier* whose exponent $c_0$ increases the influence of teacher quality; the remaining coefficients $(A', B', \alpha', \beta', \gamma')$ govern the student scaling term $h(N_S, D_S)$, controlling how model and data size reduce the distilled loss.

### CORE ARGUMENT

Distillation adds **teacher–student interaction** terms that do not appear in supervised learning:

- **Supervised ranking:** depends only on $\tilde{L}_S$ (student supervised loss).
- **Distillation ranking:** depends on $f(L_T, \tilde{L}_S, \text{capacity gap})$, i.e. contains teacher-dependent factors.

Because the capacity-gap factor $L_T/\tilde{L}_S$ appears with a negative exponent in the empirical fit, metrics that depend only on $\tilde{L}_S$ cannot predict distillation rankings in general.

### EXPLICIT FORM OF THE SCALING LAW

We adopt the empirical decomposition used in the paper:

$$L_S \propto g\left(\frac{L_T}{\tilde{L}_S}\right) h(N_S, D_S),$$

with the fitted forms

$$g\left(\frac{L_T}{\tilde{L}_S}\right) = \frac{1}{L_T^{c_0}}\left(1 + \frac{L_T}{\tilde{L}_S \cdot d_1}\right)^{-c_1 f_1}, \qquad h(N_S, D_S) = \left(\frac{A'}{N_S^{\alpha'}} + \frac{B'}{D_S^{\beta'}}\right)^{\gamma'}. \tag{13}$$

Empirical ranges used/observed (refer Busbridge et al., 2025) : $c_0 \approx 0.5$, $c_1 f_1 \approx 0.75$ (so exponent $-0.75$), $d_1 \approx 1$, $\alpha', \beta' \in [0.3, 0.5]$, $\gamma' \in [0.5, 1.0]$.

A PLAUSIBLE COUNTEREXAMPLE

**Setup:**
$$L_T = 0.10, \qquad \tilde{L}_{S_1} = 0.15 \text{ (strong)}, \qquad \tilde{L}_{S_2} = 0.25 \text{ (weak)}.$$

Using the $g(\cdot)$ form in equation 13 with $c_0 = 0.5$, $c_1 = 1.5$, $f_1 = 0.5$, $d_1 = 1$:

**Compute the teacher-amplifier factor**
$$\frac{1}{L_T^{c_0}} = \frac{1}{0.10^{0.5}} = \frac{1}{\sqrt{0.10}} = \frac{1}{0.3162277660} \approx 3.162277660.$$

**Student 1 ($\tilde{L}_{S_1} = 0.15$):**
$$r_1 = \frac{L_T}{\tilde{L}_{S_1}} = \frac{0.10}{0.15} = 0.6666666667,$$
$$1 + r_1 = 1.6666666667, \quad \ln(1.6666666667) \approx 0.5108256238,$$
$$\text{exponent} = -c_1 f_1 = -0.75$$
$$(1 + r_1)^{-0.75} = \exp\left(-0.75 \cdot 0.5108256238\right)$$
$$= \exp(-0.3831192179) \approx 0.681702.$$
Thus
$$g(r_1) \approx 3.162277660 \times 0.681702 \approx 2.155.$$

**Student 2 ($\tilde{L}_{S_2} = 0.25$):**
$$r_2 = \frac{L_T}{\tilde{L}_{S_2}} = \frac{0.10}{0.25} = 0.4,$$
$$1 + r_2 = 1.4, \quad \ln(1.4) \approx 0.3364722366,$$
$$(1 + r_2)^{-0.75} = \exp\left(-0.75 \cdot 0.3364722366\right) = \exp(-0.2523541775) \approx 0.7774,$$
so
$$g(r_2) \approx 3.162277660 \times 0.7774 \approx 2.458.$$

**Interpretation (with explicit ranking reversal).** From the computed values
$$g(r_1) \approx 2.155, \qquad g(r_2) \approx 2.458,$$
we observe that $g(r_2) > g(r_1)$, so if the student-scaling factor $h(N_S, D_S)$ were identical for the two students, then
$$L_{S_2} = g(r_2)h > g(r_1)h = L_{S_1},$$
and no ranking reversal would occur.

However, the scaling law is *multiplicative*:
$$L_S \propto g\left(\frac{L_T}{\tilde{L}_S}\right) h(N_S, D_S),$$
so a reversal can arise whenever the weaker student enjoys a sufficiently smaller scaling penalty. To make this explicit, choose
$$h_1 = 0.10, \qquad h_2 = 0.05,$$
corresponding to $S_1$ and $S_2$ student models respectively. Then the distilled losses become
$$L_{S_1} \propto g(r_1)\, h_1 = 2.155 \times 0.10 = 0.2155,$$
$$L_{S_2} \propto g(r_2)\, h_2 = 2.458 \times 0.05 = 0.1229.$$
Thus,
$$L_{S_1} > L_{S_2},$$

even though $\tilde{L}_{S_1} < \tilde{L}_{S_2}$ in supervised learning. This concrete example demonstrates the theorem's inequality:

$$\frac{g(L_T/\tilde{L}_{S_1})}{g(L_T/\tilde{L}_{S_2})} < \frac{h(N_{S_2}, D_{S_2})}{h(N_{S_1}, D_{S_1})},$$

since here

$$\frac{g(r_1)}{g(r_2)} \approx 0.877 \quad \text{and} \quad \frac{h_2}{h_1} = 0.5 < 0.877.$$

Therefore the weaker student $S_2$ becomes better than $S_1$ under distillation, illustrating explicitly how capacity-gap effects ($g$) and scaling effects ($h$) can combine to invert supervised rankings.

**Theorem 1** (Ranking divergence). *Given students $S_1, S_2$ with $\tilde{L}_{S_1} < \tilde{L}_{S_2}$, a distillation ranking reversal ($L_{S_1} > L_{S_2}$) occurs whenever*

$$\frac{g(L_T/\tilde{L}_{S_1})}{g(L_T/\tilde{L}_{S_2})} < \frac{h(N_{S_2}, D_{S_2})}{h(N_{S_1}, D_{S_1})}.$$

*Proof sketch.* From $L_S \propto g(L_T/\tilde{L}_S)\, h(N_S, D_S)$, the inequality $L_{S_1} > L_{S_2}$ is equivalent to

$$g(L_T/\tilde{L}_{S_1})\, h(N_{S_1}, D_{S_1}) > g(L_T/\tilde{L}_{S_2})\, h(N_{S_2}, D_{S_2}).$$

Since $\tilde{L}_{S_1} < \tilde{L}_{S_2}$ implies $L_T/\tilde{L}_{S_1} > L_T/\tilde{L}_{S_2}$ and $g(\cdot)$ is decreasing (negative exponent), we have $g(L_T/\tilde{L}_{S_1}) < g(L_T/\tilde{L}_{S_2})$. Thus reversal requires the $h$-ratio on the RHS to compensate for the $g$-disadvantage, giving the displayed condition. $\square$

ANALYSIS OF $g(\cdot)$ AND $h(\cdot)$

**Capacity-gap function $g(\cdot)$**

- **Teacher amplifier:** $1/L_T^{c_0}$ increases influence of teacher quality.
- **Gap penalty:** $(1 + L_T/(\tilde{L}_S d_1))^{-c_1 f_1}$ decreases with $L_T/\tilde{L}_S$ (negative exponent).
- **Behavior:** small gaps ($r \to 0$) give minimal extra benefit; intermediate gaps give a sweet spot; very large gaps are heavily penalized. Empirical optimal range $r \in [0.3, 0.6]$ (Busbridge et al., 2025).

**Student-scaling function $h(\cdot)$**

- Captures model- and data-size effects: decreasing in $N_S$ and $D_S$ (standard neural scaling).
- Teacher-independent: two students with identical $(N_S, D_S)$ have similar $h$ but can differ in $g$ due to different $\tilde{L}_S$.

## A.4 ADDITIONAL EXPERIMENTAL DETAILS

### A.4.1 KNOWLEDGE DISTILLATION BASED FINE-TUNING RESULTS

We conduct experiments on CIFAR100 (Krizhevsky et al., 2009), Caltech101 (Fei-Fei et al., 2004), and SUN397 (Xiao et al., 2010) datasets. Following prior works (Shao et al., 2022; Gholami et al., 2023), we use SGD optimizer with cosine decay scheduling and extensive hyperparameter sweeps. The projection dimension $k$ is fixed at 10240, temperature at 2, using logit-based KD with KL divergence on Nvidia L40 GPU.

**Knowledge distillation based fine-tuning details.** The ground truth for selecting or ranking pre-trained student models from a pool of candidates for a given teacher model is established by performing knowledge distillation-based fine-tuning for each teacher-student pair, using a hyperparameter sweep on the target distillation dataset. Given a teacher model and a target distillation dataset, we perform a comprehensive hyperparameter sweep over the two most critical parameters for fine-tuning: learning rate and weight decay. To constrain the search space, the temperature parameter in the KD setup is fixed at 2 across all experiments. The ground truth fine-tuning accuracy for each teacher-student pair is defined as the best performance achieved across the explored hyperparameter

configurations. All the experiments are done using a fixed seed value on a Nvidia L40 48GB vRAM GPU card with a batch size of 64. All the images were resized to 224x224 and other data augmentation strategies follow the previous transferability estimation works(Shao et al., 2022; Gholami et al., 2023). We fine-tune the models for 100 epochs for CIFAR100 and Caltech101 datasets and 50 epochs for the SUN397 image classification dataset. For reference, all the ground truth accuracies for all the 3 datasets are provided in Tables 23, 24, and 25. As for the data splits are concerned, we again follow (Shao et al., 2022) staying consistent with the previous works for fair comparisons. To be precise, for the SUN397 and the CIFAR100 image classification datasets, we use the official splits and for the Caltech101 dataset, we use random 30 images per class as trainset and the rest as test set.

As for the object detection task, we use the PASCAL VOC2012 dataset with the YOLO-v11s model as the student model (not COCO pretrained). As explained in Section 5.1, we get 12 diverse pre-trained student models. We use the official training configuration as in the Ultralytics YOLO documentation. **Teacher models training.** We use diverse teachers with varied architectures and pre-trained on different datasets. For the classification task, we use ImageNetV1 pre-trained ConvNext-Large and MaxViT-Small architectures as teachers alongwith ImageNetV2 pretrained Wide-ResNet101 and OpenCLIP based ViT-B/32 pretrained on the LAION-2B dataset(Schuhmann et al., 2022). These models are fine-tuned on the downstream target distillation dataset for 50 epochs and the corresponding accuracies are as shown in Table 22. For the ConvNext-Large, Wide-ResNet101, and MaxViT-Small teacher models, all layers were fine-tuned during training, whereas for the OpenCLIP ViT-B/32 model, only the final classification layer was fine-tuned. As for the object detection task, we use the YOLOv11L and the YOLOv11m as techer models. All of them are COCO-pretrained while some of them are additionally trained on HomeObjects-3k, and the African-Wildlife datasets to ensure diversity. These models are fine-tuned on the downstream target distillation set for 100 epochs using the official configuration settings.

**C-MoNA implementation details.** C-MoNA requires a forward pass and a backward pass to obtain the kernel matrix. Therefore, we use the Nvidia L40 48 GB vRAM GPU with batch size being 16. We keep the random projection dimension to be 10240. Our choice of $k$ is conservatively large relative to the JL bound, ensuring high-quality gradient inner product approximation while providing an excellent accuracy-computational cost tradeoff across all evaluated datasets. We adopted a unified $k = 10240$ across datasets to ensure consistent approximation quality, maintain computational efficiency, and avoid dataset-specific tuning requirements. If required, $k$ can be further tuned based on specific error tolerance requirements if computational constraints permit. To ensure that there is computationally low-overhead costs, we calculate the projected gradients for the fixed teacher once and store it. We use these stored projected gradients to calculate the cross-model NTK with each of the candidate student models.

## A.5 C-MoNA Algorithm

**Implementation Considerations.** In practice, drawing the entire $R_t \in \mathbb{R}^{k \times d_t}, R_s \in \mathbb{R}^{k \times d_s}$ can lead to memory overflow when the model parameters is large. We therefore implement a chunked processing strategy where gradients are computed and projected in batches, with matrices grown dynamically through concatenation. The FastJL library provides computational acceleration for the random projections and crucially ensures that both $R_t$ and $R_s$ are drawn from identical Rademacher distributions using the same random seed. This guarantees that the cross-model NTK comparisons remain theoretically sound, as the projection preserves relative inner product structures between teacher and student gradient spaces under the Johnson-Lindenstrauss lemma.

This approach balances computational efficiency with theoretical rigor while making the algorithm scalable to large model architectures.

## A.6 C-MoNA's consistent performance across diverse teacher models

Fig. 2a clearly shows that C-MoNA demonstrates exceptional robustness across diverse experimental conditions, achieving remarkably low standard deviations (0.044 average) compared to competing methods (0.122–0.216 range) in full trainset settings. The method maintains superior consistency across three datasets (SUN397, CIFAR100, Caltech101). Notably, C-MoNA exhibits minimal performance variation across heterogeneous teacher models spanning CNN-only (WideResNet101), CNN-Transformer hybrid (MaxViTSmall), and Transformer-only (ViT-B/32) architectures,

---

**Algorithm 1:** C-MoNA: Cross-Model NTK Alignment for Student Selection

---

**Input:** Teacher $f_t(\cdot; \theta_t)$;    Students $\{f_{s_m}(\cdot; \theta_{s_m})\}_{m=1}^M$;    Unlabeled set $\mathcal{D} = \{x_n\}_{n=1}^N$;
     Projection dim. $k$;    Param. dims. $d_t, d_{s_m}$
**Output:** Students ranked by C-MoNA scores $\{\alpha_m\}_{m=1}^M$
Initialize $G_t \in \mathbb{R}^{k \times N}$ and each $G_{s_m} \in \mathbb{R}^{k \times N}$      `// storage for projected`
   `gradients`
Draw $R_t \in \mathbb{R}^{k \times d_t}, R_s \in \mathbb{R}^{k \times d_{s_m}}$ from same underlying Rademacher distribution with fixed
   seed                       `// same distribution, separate matrices`
**for** $n \leftarrow 1$ **to** $N$ **do**
     $g_t \leftarrow \nabla_{\theta_t} f_t(x_n; \theta_t)$                     `// teacher gradient in` $\mathbb{R}^{d_t}$
     $\tilde{g}_t \leftarrow R_t\, g_t$                            `// projected to` $\mathbb{R}^k$ `using` $R_t$
     set column $n$ of $G_t \leftarrow \tilde{g}_t$
     **for** $m \leftarrow 1$ **to** $M$ **do**
         $g_{s_m} \leftarrow \nabla_{\theta_{s_m}} f_{s_m}(x_n; \theta_{s_m})$         `// student gradient in` $\mathbb{R}^{d_{s_m}}$
         $\tilde{g}_{s_m} \leftarrow R_s\, g_{s_m}$              `// projected gradient using` $R_s$
         set column $n$ of $G_{s_m} \leftarrow \tilde{g}_{s_m}$
     **end**
**end**
**for** $m \leftarrow 1$ **to** $M$ **do**
     $K_{t,s_m} \leftarrow G_t\, G_{s_m}^\top$                       `// cross-NTK Gram matrix`
     $K_{t,t} \leftarrow G_t\, G_t^\top,$    $K_{s_m,s_m} \leftarrow G_{s_m}\, G_{s_m}^\top$           `// self-NTKs`
     $\alpha_m \leftarrow \|K_{t,s_m}\|_F / \sqrt{\|K_{t,t}\|_F\, \|K_{s_m,s_m}\|_F}$       `// compatibility score`
**end**
**return** Students sorted by descending $\alpha_m$         `// best student first`

---

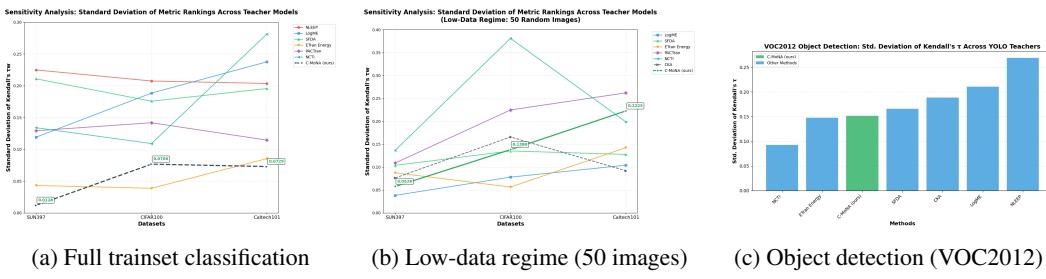

      (a) Full trainset classification       (b) Low-data regime (50 images)       (c) Object detection (VOC2012)

Figure 2: Standard deviation of Kendall's $\tau$ across four diverse teacher models in different experimental settings. C-MoNA consistently demonstrates the lowest variability across (a) full trainset image classification, (b) low-data regime classification with 50 random images, and (c) object detection tasks on VOC2012 with 100 random images.

demonstrating architectural agnosticism that traditional methods lack. This cross-teacher consistency validates that C-MoNA's gradient-space alignment captures fundamental compatibility patterns independent of specific architectural choices or pre-training objectives. Critically, as shown in Fig. 2b, C-MoNA's advantages amplify in low-data regimes (50 random images), where traditional transferability metrics suffer numerical instabilities or complete failures, while C-MoNA continues providing reliable estimations. This cross-model NTK alignment approach proves fundamentally more robust than model-data relationship methods, enabling confident student selection across varying data availability constraints. The comprehensive analysis establishes C-MoNA as the most reliable solution for practical knowledge distillation pipelines in both data-rich and privacy-constrained scenarios. A similar inference can be drawn from the object detection sensitivity plot Fig. 2c as well. C-MoNA demonstrates superior performance in the object detection domain, achieving both the highest average Kendall's tau and one of the modest standard deviation across four diverse YOLO teacher models. C-MoNA, being an unsupervised metric, maintains stable positive correlations across all teacher variants. This dual advantage of accuracy and consistency validates that cross-model NTK alignment

successfully generalizes beyond image classification to object detection tasks, providing reliable student selection capabilities across heterogeneous detection architectures trained on diverse datasets.

### A.6.1 Evaluating CKA and alternative C-MoNA Matrix Reduction Approaches

Table 9 clearly shows that C-MoNA gains 3.3x over CKA across 4 diverse teachers averaged across 3 image classification datasets empirically supporting the theoretical reasoning in Section 4.3. Tables 10 and 11 present the performance of different C-MoNA matrix reduction strategies on the object detection task. The results demonstrate that our proposed approach of utilizing all cross-covariance values provides both theoretical rigor and empirical robustness. Alternative reduction methods, such as trace-based reduction or selecting only the top-10% energies (spectral), show suboptimal performance when evaluated across multiple teachers and experimental configurations.

While trace-based reduction may offer some utility in constrained scenarios—particularly when diverse samples are unavailable or when working with very limited similar samples that result in unfavorable signal-to-noise ratios from cross-covariance values—it is not suitable for general applications. For most practical use cases, we advocate for our theoretically-grounded and empirically-validated reduction strategy, which incorporates all cross-covariance values in the estimation computation, thereby providing more comprehensive and reliable performance.

Table 9: Comparison of the transferability estimation performance (Kendall's tau,$\tau_w$) of naive implementation of kernel alignment compared to C-MoNA averaged across 4 teachers for a random 50 images subset for the classification task.

|  | CIFAR100 | SUN397 | Caltech101 | Avg |
|---|---|---|---|---|
| CKA | -0.009 | -0.171 | -0.283 | -0.154 |
| C-MoNA (ours) | 0.465 | 0.29 | 0.313 | 0.356 |

Table 10: Kendall's tau ($\tau_w$) for the different C-MoNA matrix reduction strategies shown for the object detection task on the VOC2012 dataset random 100 images subset."frob" is frobenius norm proposed.

|  | Yolo11l-Coco | Yolo11l-Home | Yolo11m-Home | Yolo11m-Wild | Avg. |
|---|---|---|---|---|---|
| CKA | 0.037 | -0.152 | -0.428 | -0.391 | -0.233 |
| Spectral | -0.171 | 0.033 | 0.262 | -0.112 | 0.003 |
| Trace | -0.435 | 0.146 | **0.581** | **0.410** | 0.170 |
| C-MoNA(frob) | **0.509** | **0.236** | 0.421 | 0.122 | **0.322** |

Table 11: Kendall's tau ($\tau_w$) for the different C-MoNA matrix reduction strategies shown for the object detection task on the VOC2012 dataset random 25 images subset. "frob" is frobenius norm proposed.

|  | Yolo11l-Coco | Yolo11l-Home | Yolo11m-Home | Yolo11m-Wild | Avg. |
|---|---|---|---|---|---|
| CKA | -0.066 | -0.014 | -0.427 | -0.245 | -0.188 |
| Spectral | -0.228 | -0.185 | 0.147 | **0.429** | 0.04 |
| Trace | -0.435 | 0.028 | **0.470** | 0.263 | 0.081 |
| C-MoNA(frob) | **0.257** | **0.297** | 0.293 | 0.272 | **0.279** |

### A.7 Additional Experimental Results

Table 6 report the average performance across all the teacher models in low-data regimes. Tables 12, 13, and 14 gives the detailed per teacher results for each of the low data setting including the CKA performance evaluation on these settings for the Caltech-101 dataset. Tables 15, 16, 17, and 18 gives the detailed per teacher results for each of the low data setting including the CKA performance evaluation on these settings for the SUN397 dataset. Tables 19, 20, and 21 gives the detailed per

teacher results for each of the low data setting including the CKA performance evaluation on these settings for the CIFAR-100 dataset. Tables 22, 23, 24, and 25 present the performance of teacher models and the post-distillation accuracies respectively, which serve as ground-truth values for computing the Weighted Kendall's tau ($\tau_w$) metric. This metric is used to evaluate the predictive estimation capabilities of all benchmarked metrics in the image classification task. What sets C-MoNA apart from most existing transferability estimation methods is its ability to enable model selection in low-data regimes. Notably, C-MoNA demonstrates strong performance even when using just a small number of randomly sampled images. Given that even 25 randomly selected images already deliver impressive results compared to existing benchmarks, we argue that a carefully curated set of 25 representative images could lead to even more reliable performance with C-MoNA—further underscoring its practicality and robustness in low-data settings. All the experimental results clearly showcase the robustness, consistent superior performance of the C-MoNA metric across all datasets, experimental configurations, and diverse pre-trained teachers. Along with the strong performance and reliability of C-MoNA, it is simple to implement and easy to scale to larger architectures making it an easy-to-deploy metric that would positively benefit the distillation performance. To analyze the sensitivity analyses across projection dimensions, we have Fig. 3. These plots present both mean ± standard deviation and 95% confidence intervals. For the projection dimension, the theoretical minimum is defined by the Johnson-Lindenstrauss lemma as $k = O\left(\frac{\log n}{\epsilon^2}\right)$, where $k$ is the projection dimension, $n$ is the number of samples, and $\epsilon$ represents the distortion error.

Fig. 3 validates this theoretical bound empirically. As can be seen from Fig. 3a and Fig. 3b, in the 50-sample regime, performance remains stable across projection dimensions, the difference between the maximum dimension (10,240) and minimum dimension (1,024) is negligible, consistent with the JL lemma's prediction. However, when increasing to 1,000 samples, the impact of projection dimension becomes more pronounced, as seen from Fig. 3c and Fig. 3d. This aligns with the JL lemma, which indicates that larger sample sizes require higher projection dimensions to maintain the same error rate. To avoid dataset-specific tuning across our experiments, we conservatively use 10,240 as our default projection dimension. Based on our sensitivity analyses in Fig. 3, we recommend a projection dimension greater than 5,000 for large sample sets. Integrating theoretical bounds with our comprehensive experimental validation (Fig. 1), we recommend 50 unlabeled samples and a projection dimension of 1,024 as practical minimums when using randomly sampled (non-curated) data. This configuration consistently achieves positive correlations and typically places at least 3 out of 5 correct models in the top-5 predictions, making it practically feasible.

Table 12: Kendall's tau ($\tau_w$) for the Caltech101 dataset using 50 random images.

|  | MaxViT-Small | Wide-ResNet101 | ConvNext-Large | OpenCLIP-ViT-B/32 | Avg. |
|---|---|---|---|---|---|
| ETran Energy | -0.208 | -0.261 | -0.512 | -0.133 | -0.159 |
| SFDA | 0.046 | -0.066 | -0.185 | **0.158** | -0.011 |
| LogME | 0.000 | 0.102 | -0.034 | -0.188 | -0.030 |
| PAC | -0.035 | **0.442** | -0.142 | -0.241 | 0.006 |
| NCTI | 0.276 | 0.314 | 0.264 | -0.172 | 0.170 |
| CKA | -0.241 | -0.432 | -0.276 | -0.184 | -0.283 |
| C-MoNA (ours) | **0.682** | 0.136 | **0.297** | 0.139 | **0.313** |

Table 13: Kendall's tau ($\tau_w$) for the Caltech101 image classification dataset using 25 random images. "-" denotes values that are NaN or unavailable due to numerical instability or insufficient samples required for computation.

|  | MaxViT-Small | Wide-ResNet101 | ConvNext-Large | OpenCLIP-ViT-B/32 | Avg. |
|---|---|---|---|---|---|
| ETran Energy | -0.156 | -0.261 | -0.519 | -0.301 | -0.309 |
| SFDA | 0.023 | -0.060 | -0.164 | **0.324** | 0.030 |
| LogME | -0.015 | 0.067 | -0.056 | -0.161 | -0.041 |
| PAC | -0.061 | 0.031 | 0.187 | -0.103 | 0.013 |
| NCTI | -0.081 | -0.236 | 0.201 | - | - |
| CKA | -0.208 | -0.271 | -0.290 | 0.104 | -0.166 |
| C-MoNA (ours) | **0.490** | **0.119** | **0.469** | 0.253 | **0.332** |

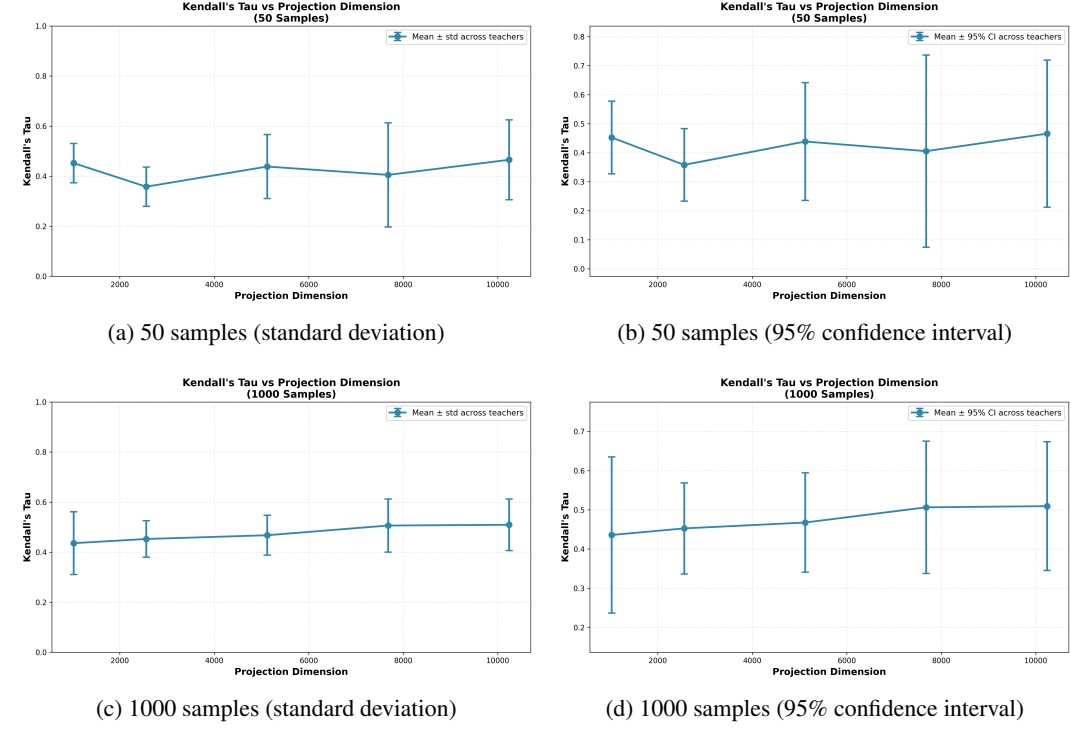

(a) 50 samples (standard deviation)

(b) 50 samples (95% confidence interval)

(c) 1000 samples (standard deviation)

(d) 1000 samples (95% confidence interval)

Figure 3: Impact of projection dimension on Kendall's $\tau_w$ correlation stability across different sample sizes. Error bars show (a,b) results with 50 CIFAR100 samples and (c,d) results with 1000 CIFAR100 samples, represented as standard deviation and 95% confidence intervals respectively. The metric exhibits consistent performance across projection dimensions from 1024 to 10240, with reduced variance in the high-sample regime.

Table 14: Kendall's tau ($\tau_w$) for the Caltech101 dataset using 10 random images.

|  | MaxViT-Small | Wide-ResNet101 | ConvNext-Large | OpenCLIP-ViT-B/32 | Avg. |
|---|---|---|---|---|---|
| ETran Energy | -0.272 | -0.261 | -0.499 | -0.271 | -0.325 |
| LogME | -0.015 | -0.004 | -0.056 | -0.122 | -0.049 |
| PAC | -0.005 | -0.020 | -0.221 | -0.193 | -0.109 |
| CKA | -0.088 | -0.470 | -0.470 | -0.249 | -0.319 |
| C-MoNA (ours) | **0.138** | **0.058** | **0.232** | **0.210** | **0.159** |

Table 15: Kendall's tau ($\tau_w$) using 100 random samples per teacher model on SUN397 dataset.

|  | OpenCLIP ViT-B/32 | ConvNeXt-Large | MaxViT-Small | Wide-ResNet101 | Avg. |
|---|---|---|---|---|---|
| ETran Energy | -0.225 | -0.325 | -0.339 | -0.249 | -0.284 |
| SFDA | **0.268** | 0.128 | -0.039 | 0.353 | 0.177 |
| LogME | -0.192 | -0.175 | -0.181 | -0.214 | -0.190 |
| PAC | -0.409 | -0.245 | -0.049 | -0.354 | -0.264 |
| NCTI | 0.072 | 0.041 | -0.369 | **0.406** | 0.037 |
| CKA | 0.195 | 0.190 | 0.141 | 0.102 | 0.157 |
| C-MoNA (ours) | 0.266 | **0.278** | **0.391** | 0.250 | **0.296** |

Table 16: Kendall's tau ($\tau_w$) using 50 random samples per teacher model on SUN397 image classification dataset. "-" denotes values that are NaN due to numerical instability.

|  | OpenCLIP ViT-B/32 | ConvNeXt-Large | MaxViT-Small | Wide-ResNet101 | Avg. |
|---|---|---|---|---|---|
| ETran Energy | -0.182 | -0.399 | -0.353 | -0.234 | -0.292 |
| SFDA | **0.297** | 0.105 | 0.072 | **0.292** | 0.191 |
| LogME | -0.192 | -0.156 | -0.196 | -0.101 | -0.161 |
| PAC | -0.363 | -0.131 | -0.090 | -0.268 | -0.213 |
| NCTI | 0.016 | -0.293 | -0.028 | - | - |
| CKA | -0.303 | -0.128 | -0.129 | -0.124 | -0.171 |
| C-MoNA (ours) | 0.239 | **0.386** | **0.253** | 0.282 | **0.290** |

Table 17: Kendall's tau ($\tau_w$) using 25 random samples per teacher model on SUN397 image classification dataset. "–" denotes values that are NaN due to numerical instability.

|  | OpenCLIP ViT-B/32 | ConvNeXt-Large | MaxViT-Small | Wide-ResNet101 | Avg. |
|---|---|---|---|---|---|
| ETran Energy | -0.201 | -0.347 | -0.393 | -0.236 | -0.294 |
| SFDA | – | – | – | – | – |
| LogME | -0.186 | -0.156 | -0.238 | -0.214 | -0.198 |
| PAC | -0.008 | 0.118 | 0.097 | -0.002 | 0.051 |
| NCTI | – | – | – | – | – |
| CKA | -0.352 | -0.195 | -0.184 | -0.289 | -0.255 |
| C-MoNA (ours) | **0.325** | **0.499** | **0.435** | **0.205** | **0.366** |

Table 18: Kendall's tau ($\tau_w$) using 10 random samples per teacher model on SUN397 dataset.

|  | OpenCLIP ViT-B/32 | ConvNeXt-Large | MaxViT-Small | Wide-ResNet101 | Avg. |
|---|---|---|---|---|---|
| ETran Energy | -0.227 | -0.298 | -0.451 | -0.230 | -0.301 |
| LogME | -0.192 | -0.033 | -0.216 | -0.183 | -0.156 |
| PAC | **0.168** | -0.208 | -0.272 | 0.064 | -0.062 |
| CKA | -0.196 | -0.465 | -0.533 | -0.341 | -0.383 |
| C-MoNA (ours) | 0.150 | **-0.074** | **-0.010** | **0.170** | **0.059** |

Table 19: Kendall's tau ($\tau_w$) for the CIFAR-100 dataset using 50 random images.

|  | Wide-ResNet101 | ConvNexT-Large | MaxViT-Small | OpenCLIP-ViT-B/32 | Avg. |
|---|---|---|---|---|---|
| ETran Energy | -0.168 | -0.208 | -0.055 | -0.156 | -0.146 |
| SFDA | 0.144 | 0.502 | 0.386 | **0.252** | 0.321 |
| LogME | -0.269 | -0.375 | -0.322 | -0.164 | -0.282 |
| PAC | -0.160 | -0.504 | -0.317 | 0.109 | -0.218 |
| NCTI | -0.564 | 0.369 | 0.093 | -0.436 | -0.134 |
| CKA | -0.068 | 0.254 | -0.024 | -0.201 | -0.009 |
| C-MoNA (ours) | **0.612** | **0.516** | **0.495** | 0.239 | **0.465** |

Table 20: Kendall's tau ($\tau_w$) for CIFAR-100 image classification dataset using 25 random images.

| Metric | Wide-ResNet101 | ConvNeXT-Large | MaxViT-Small | OpenCLIP-ViT-B/32 | Avg. |
|---|---|---|---|---|---|
| ETran Energy | -0.257 | -0.159 | -0.020 | -0.239 | -0.168 |
| SFDA | 0.250 | 0.580 | 0.402 | 0.161 | 0.348 |
| LogME | -0.248 | -0.439 | -0.305 | -0.234 | -0.306 |
| PAC | -0.006 | -0.234 | -0.243 | -0.093 | -0.144 |
| NCTI | -0.192 | 0.143 | -0.391 | -0.418 | -0.214 |
| CKA | -0.114 | 0.186 | 0.245 | 0.002 | 0.079 |
| C-MoNA (ours) | **0.312** | **0.682** | **0.628** | **0.186** | **0.452** |

Table 21: Kendall's tau ($\tau_w$) for the CIFAR-100 image classification dataset using 10 random images. "–" denotes values that are NaN or unavailable due to numerical instability or insufficient samples required for computation.

| Metric | Wide-ResNet101 | ConvNeXT-Large | MaxViT-Small | OpenCLIP-ViT-B/32 | Avg. |
|---|---|---|---|---|---|
| ETran Energy | -0.154 | -0.094 | -0.021 | -0.161 | -0.107 |
| SFDA | – | 0.454 | – | – | – |
| LogME | -0.113 | -0.391 | -0.222 | -0.027 | -0.188 |
| PAC | -0.117 | -0.256 | -0.379 | -0.287 | -0.259 |
| NCTI | – | -0.284 | – | – | – |
| CKA | -0.232 | -0.120 | -0.140 | -0.028 | -0.13 |
| C-MoNA (ours) | **0.513** | **0.502** | **0.345** | **0.299** | **0.414** |

Table 22: Accuracy of various teacher models across 3 different image classification datasets.

| | SUN397 | CIFAR100 | Caltech101 |
|---|---|---|---|
| Wide-ResNet101 | 66.12 | 87.57 | 93.36 |
| ConvNeXT-Large | 70.45 | 90.01 | 94.94 |
| MaxViT-Small | 69.65 | 90.82 | 94.55 |
| OpenCLIP-ViT-B/32 | 75.11 | 84.50 | 93.12 |

Table 23: Knowledge distillation based fine-tuning accuracy of all teacher–student pairs for the SUN397 image classification dataset.

| Students ↓ \ Teachers → | Wide-ResNet101 | ConvNext-Large | MaxViT-Small | OpenCLIP ViT-B/32 |
|---|---|---|---|---|
| ResNet34 | 62.27 | 64.23 | 64.29 | 62.52 |
| MobileNetV2 | 60.74 | 62.65 | 62.39 | 60.24 |
| ShuffleNetV2x1.5 | 60.93 | 63.15 | 63.12 | 60.20 |
| EfficientNet-B0 | 63.24 | 66.25 | 66.05 | 63.38 |
| ResNet18 | 60.18 | 62.58 | 61.74 | 59.75 |
| DeiT-T/16 | 62.01 | 63.30 | 63.15 | 62.19 |
| DeiT-T/16 (Distilled) | 62.53 | 63.58 | 63.45 | 63.15 |
| GoogleNet | 60.02 | 63.09 | 62.49 | 58.83 |
| SwinT-Tiny | 66.74 | 68.47 | 68.30 | 67.24 |
| MnasNet | 61.55 | 64.76 | 63.72 | 61.41 |
| DeiT-S/16 | 65.50 | 67.47 | 67.78 | 66.54 |
| DeiT-S/16 (Distilled) | 65.89 | 67.98 | 68.27 | 67.06 |

Table 24: Knowledge distillation based fine-tuning accuracy of all teacher–student pairs for the CIFAR100 image classification dataset.

| Students ↓ \ Teachers → | Wide-ResNet101 | ConvNext-Large | MaxViT-Small | OpenCLIP ViT-B/32 |
|---|---|---|---|---|
| ResNet34 | 83.59 | 85.31 | 85.93 | 82.15 |
| MobileNetV2 | 81.41 | 82.60 | 83.13 | 79.85 |
| ShuffleNetV2x1.5 | 82.60 | 84.51 | 82.88 | 80.75 |
| EfficientNet-B0 | 86.17 | 87.22 | 86.36 | 83.55 |
| ResNet18 | 82.47 | 83.74 | 83.54 | 80.53 |
| DeiT-T/16 | 82.88 | 85.35 | 84.72 | 81.56 |
| DeiT-T/16 (Distilled) | 84.27 | 85.77 | 85.29 | 82.04 |
| GoogleNet | 82.14 | 84.51 | 83.52 | 80.73 |
| SwinT-Tiny | 87.82 | 88.17 | 88.37 | 84.25 |
| MnasNet | 83.69 | 85.17 | 84.43 | 82.24 |
| DeiT-S/16 | 87.16 | 88.28 | 88.28 | 83.75 |
| DeiT-S/16 (Distilled) | 86.90 | 88.38 | 88.49 | 84.07 |

Table 25: KD-based fine-tuning accuracy of all teacher–student pairs for Caltech101.

| Students ↓ \ Teachers → | Wide-ResNet101 | ConvNext-Large | MaxViT-Small | OpenCLIP ViT-B/32 |
|---|---|---|---|---|
| ResNet34 | 92.40 | 92.34 | 92.45 | 91.36 |
| MobileNetV2 | 90.25 | 90.25 | 90.06 | 89.54 |
| ShuffleNetV2x1.5 | 89.77 | 91.27 | 87.03 | 88.81 |
| EfficientNet-B0 | 92.86 | 92.67 | 92.87 | 91.60 |
| ResNet18 | 88.27 | 90.21 | 86.92 | 89.91 |
| DeiT-T/16 | 89.67 | 90.25 | 90.31 | 89.17 |
| DeiT-T/16 (Distilled) | 89.70 | 90.02 | 90.71 | 89.49 |
| GoogleNet | 90.30 | 91.34 | 91.96 | 90.08 |
| SwinT-Tiny | 93.04 | 92.85 | 93.17 | 91.97 |
| MnasNet | 89.31 | 90.74 | 89.28 | 89.26 |
| DeiT-S/16 | 92.63 | 92.60 | 92.50 | 91.63 |
| DeiT-S/16 (Distilled) | 92.75 | 92.77 | 92.72 | 92.08 |

Table 26: Task performance when using the selected models for the best two methods post KD fine-tuning on CIFAR100 dataset across different teacher models.

| Method | MaxViT-Small | WideResNet-101 | ConvNeXt-Large | OpenCLIP-ViT-B/32 | Mean±Std |
|---|---|---|---|---|---|
| NLEEP | 88.28 | 83.69 | 85.31 | 81.56 | 84.71±2.83 |
| C-MoNA (ours) | 88.28 | 87.16 | 88.28 | 83.75 | 86.87±2.14 |
| Oracle | **88.49** | **87.82** | **88.38** | **84.25** | **87.23±2.01** |

Table 27: Task performance when using the selected models for the best two methods post KD fine-tuning on Caltech101 dataset across different teacher models.

| Method | MaxViT-Small | WideResNet-101 | ConvNeXt-Large | OpenCLIP-ViT-B/32 | Mean±Std |
|---|---|---|---|---|---|
| SFDA | 90.06 | 89.70 | 92.77 | 89.26 | 90.45±1.58 |
| C-MoNA (ours) | 92.50 | 92.63 | 92.60 | 91.63 | 92.34±0.48 |
| Oracle | **93.17** | **93.04** | **92.85** | **92.08** | **92.78±0.49** |

