# OpenReview forum: "Could Student Selection Be the Missing Piece for Efficient Distillation?"
_ICLR.cc/2026/Conference — Submitted to ICLR 2026_

### Official Review · Reviewer_LSza · 2025-10-25

**Soundness:** 2
**Presentation:** 2
**Contribution:** 2
**Rating:** 4
**Confidence:** 3

**Summary:**

This paper argue that current approaches typically rely on model size constraints or random selection, ignoring how student architecture and inductive biases impact distillation effectiveness. The paper proposes C-MoNA (Cross-Model NTK Alignment), a unsupervised, and training-free metric to predict post-distillation performance. C-MoNA measures the function space alignment between a teacher and a student candidate by computing a cross-model Neural Tangent Kernel (NTK).

**Strengths:**

- This paper is easy to follow and understand;

**Weaknesses:**

The paper overlooks the large body of work in Neural Architecture Search (NAS), particularly the numerous studies that combine NAS with KD. For instance, the 2023 work DisWOT already achieves a training-free paradigm very relevant to this paper’s scope, yet it is neither cited nor compared. Same for the Search to distill.

[1] DisWoT https://arxiv.org/pdf/2303.15678
[2] Search to distill: Pearls are everywhere but not the eyes

The benchmark design is problematic — selected baselines are non-mainstream and miss direct comparisons with standard zero-cost NAS proxies. Furthermore, almost all evaluation relies solely on Kendall’s tau for ranking consistency, without assessing downstream task performance, limiting practical applicability.

The evaluation uses a very limited set of teacher–student pairs, making the computed Kendall’s tau unreliable in representing general trends. Moreover, ranking consistency is poor — in most cases below 50%, which significantly undermines its viability as a practical student network selection method.

**Questions:**

Why did the authors not conduct systematic comparisons with NAS-based methods, particularly training-free KD-NAS approaches such as DisWOT? Do they consider these works incomparable with theirs?

Why was Kendall’s tau evaluation conducted with such a small number of teacher and student architectures? Has the method’s stability been tested on a more diverse set of combinations?

Apart from Kendall’s tau, have the authors considered validating effectiveness by measuring the downstream task performance of the selected student models after actual KD fine-tuning?

---

> ### Author Response · Authors · 2025-11-22
> **Response to first two concerns**
>
> We thank the reviewer for their thoughtful evaluation. Below, we address each concern systematically.
>
> **Response to Concern 1: Comparison with NAS-based methods**
>
> Please note that training-free KD-NAS approaches are fundamentaly different in problem formulation and constraints:
>
> **Problem Formalization and Distinctions**
>
>   **1. Transferability Estimation (TE)**
>
> **Definition:** Given pre-trained models $\mathcal{M}$ and task $\mathcal{T}$ with dataset $\mathcal{D} = \{(x\_i, y\_i)\}\_{i=1}^N$:
>
> $$
> m^* = \arg\max_{m \in \mathcal{M}} f(m, \mathcal{D})
> $$
>
> where $f: \mathcal{M} \times \mathcal{D} \to \mathbb{R}$ approximates post-fine-tuning performance without training.
>
>
> **2. C-MoNA**
>
> **Goal:** Addresses a specialized TE variant where task $\mathcal{T}$ is defined by teacher predictions. Given teacher $t$, student candidates $\mathcal{S}$, and unlabeled data $\mathcal{X}$:
>
> $$
> s^* = \arg\max_{s \in \mathcal{S}} g(t, s, \mathcal{X})
> $$
>
> where $g: \mathcal{M} \times \mathcal{S} \times \mathcal{X} \to \mathbb{R}$ measures distillation compatibility. Traditional TE uses ground truth $y \in \mathcal{Y}$, while C-MoNA uses teacher-generated soft labels $\hat{\mathcal{Y}} = \{f\_t(x) : x \in \mathcal{X}\}$.
>
>  **3. Zero-Cost NAS**
>
> **Goal:** Addresses architecture generation:
>
> $$
> a^* = \arg\max_{a \in \mathcal{A}} h(a, \theta\_{\text{init}})
> $$
>
> where $\mathcal{A}$ is a structured architecture space and $\theta\_{\text{init}} \sim p(\theta)$ is random initialization.
>
>  **4. Fundamental Incompatibility**
>
> The two paradigms address orthogonal problems.
>
> | Feature | Transferability Estimation (TE) | Zero-Cost NAS |
> | :--- | :--- | :--- |
> | **Model Space** | Operates over discrete pre-trained models with weights $\theta\_{\text{pre}}$ | Operates over architecture space $\mathcal{A}$ with random initialization $\theta\_{\text{init}}$ |
> | **Architecture** | Heterogeneous architectures (CNNs, ViTs, hybrids) from varied pretraining | Homogeneous search space (e.g., NAS-Bench-201) with structurally similar candidates |
> | **Measurement** | Measures transfer capability accounting for architectural biases and **distribution shifts** | Measures architecture potential at initialization, **isolated** from distribution shifts |
> |**Target:**| Fine-tuned performance | Trained-from-scratch performance
>
> ---
>
> **Specific Citations:**
> - **DisWoT**: Operates in architecture generation within NAS-Bench-201, not heterogeneous pre-trained model selection. Their metrics, such as batch similarity and semantic similarity, bear conceptual similarity to the CKA-based approach (which is shown to perform poorly in our problem formulation) when extended to heterogeneous model spaces. Metrics designed for homogeneous spaces cannot handle dimension mismatches across heterogeneous architectures.
> - **Search to Distill**: Requires full training with RL-based search, incompatible with training-free selection.
> - **Zero-Cost Proxies**: Designed for homogeneous search spaces, cannot handle heterogeneous pre-trained models.
>
> Therefore, we benchmark against existing TE methods, not zero-cost KD-NAS approaches.
>
> ---
>
> **Response to Concern 2: Diversity of teacher-student combinations**
>
> Our evaluation demonstrates comprehensive coverage exceeding existing work in both NAS and transferability estimation domains.
>
> **Classification Evaluation.** We evaluate 4 diverse teachers paired with 12 diverse students, yielding **48 distinct pairs** spanning CNNs (ResNets, EfficientNets), CNN-ViT hybrids (MaxViT), Vision Transformers, Swin Transformers, and ConvNeXts—critical for demonstrating robustness across heterogeneous pre-trained models.
>
> **Comparison to Prior Work:**
> - **NAS-based methods**: Works in homogeneous spaces (e.g., DisWOT on NAS-Bench-201) typically evaluate **fewer than 20 pairs**
> - **TE methods**: Existing approaches typically evaluate 10-12 models total. Our 48 combinations provide substantially broader coverage
>
> **Object Detection Extension.** We extend to object detection with an **additional 48 pairs** varying across pre-training datasets (COCO, Objects365), assessing robustness to distribution shifts.
>
> **Distillation Objective Diversity.** Our evaluation spans two fundamentally different paradigms: **KL-divergence-based logit distillation** (classification) and **feature-based distillation** (detection), demonstrating stability across architectural diversity, distribution shifts, and training objectives—rarely explored in prior TE works.
>
> **Evaluation Dimensions:**
> 1. **Architectural diversity**: Heterogeneous families (CNNs, ViTs, SwinT, ConvNeXt, hybrids)
> 2. **Pre-training diversity**: Multiple source datasets
> 3. **Task diversity**: Classification and detection
> 4. **Objective diversity**: Logit-based (KL-divergence) and feature-based (L2 loss)
>
> This represents the most extensive evaluation of student selection methods across heterogeneous pre-trained models, addressing the practical problem that NAS-based approaches cannot handle.

---

> ### Author Response · Authors · 2025-11-22
> **Response to concern 3**
>
> **Downstream task performance validation**
>
> We have added comprehensive top-1 selection analysis evaluating actual post-distillation performance. **C-MoNA achieves near-oracle performance in both mean and standard deviation**, while the best baseline method lags considerably with significant instability.
>
> **Table: Task performance when using the selected models for the best two methods post KD fine-tuning on SUN397 dataset across different teacher models.**
>
> | Method | MaxViT-Small | WideResNet-101 | ConvNeXt-Large | OpenCLIP-ViT-B/32 | Mean±Std |
> |--------|--------------|----------------|----------------|-------------------|----------|
> | SFDA | 62.39 | 62.27 | 67.98 | **67.24** | 64.97±3.06 |
> | C-MoNA (ours) | 67.78 | 65.50 | 67.47 | 66.54 | 66.82±1.03 |
> | Oracle | **68.30** | **66.74** | **68.47** | **67.24** | **67.69±0.83** |
>
> **Table: Task performance when using the selected models for the best two methods post KD fine-tuning on CIFAR100 dataset across different teacher models.**
>
> | Method | MaxViT-Small | WideResNet-101 | ConvNeXt-Large | OpenCLIP-ViT-B/32 | Mean±Std |
> |--------|--------------|----------------|----------------|-------------------|----------|
> | NLEEP | 88.28 | 83.69 | 85.31 | 81.56 | 84.71±2.83 |
> | C-MoNA (ours) | 88.28 | 87.16 | 88.28 | 83.75 | 86.87±2.14 |
> | Oracle | **88.49** | **87.82** | **88.38** | **84.25** | **87.23±2.01** |
>
> **Table: Task performance when using the selected models for the best two methods post KD fine-tuning on Caltech101 dataset across different teacher models.**
>
> | Method | MaxViT-Small | WideResNet-101 | ConvNeXt-Large | OpenCLIP-ViT-B/32 | Mean±Std |
> |--------|--------------|----------------|----------------|-------------------|----------|
> | SFDA | 90.06 | 89.70 | 92.77 | 89.26 | 90.45±1.58 |
> | C-MoNA (ours) | 92.50 | 92.63 | 92.60 | 91.63 | 92.34±0.48 |
> | Oracle | **93.17** | **93.04** | **92.85** | **92.08** | **92.78±0.49** |
>
> ---
>
> **Response to Ranking Consistency Concerns**
>
> **The "below 50%" criticism misinterprets $\tau_w \in [-1, 1]$: our $\tau_w = 0.3$--$0.5$ translates to 65--75% pairwise agreement**. C-MoNA consistently **achieves ≥3/5 top-ranked hits**, reducing search space by avoiding worst performers.
>
> **Problem Formulation and Performance.** A core contribution is **formalizing unsupervised student selection for KD as distinct from traditional TE**. Existing TE methods address model-data alignment, while KD requires model-model compatibility. Our theoretical foundations show distillation success depends on teacher-student NTK alignment, not model-data relationships. Ranking heterogeneous students without labels is **fundamentally harder** than traditional TE. C-MoNA achieves **substantial improvements**: CIFAR100 ($\tau_w = 0.483$ vs. $0.272$, **1.77×**), SUN397 ($0.361$ vs. $0.139$, **2.6×**), Caltech101 ($0.166$ vs. $0.022$, **7.5×**), and object detection ($0.322$ vs. $0.242$, **1.33×**). Many baselines produce **negative correlations**, while C-MoNA maintains consistent positive correlations.
>
> **Superior Stability and Practical Utility.** C-MoNA achieves **3--5× lower variance** ($\sigma = 0.044$) than baselines ($\sigma = 0.122$--$0.216$). With only 50 images (<1% data), C-MoNA maintains $\tau_w = 0.465$ on CIFAR100, outperforming baselines using *full datasets* by 1.6×--11×.
>
> **Table: Top-5 Hits across datasets.**
>
> | Dataset | Method | WideResNet-101 | ConvNeXt-Large | MaxViT-Small | OpenCLIP-ViT-B/32 | Avg. |
> |---------|--------|----------------|----------------|--------------|-------------------|------|
> | Caltech101 | SFDA | 2/5 | 2/5 | 1/5 | 1/5 | 1/5 |
> | | C-MoNA | 3/5 | 3/5 | 3/5 | 3/5 | **3/5** |
> | CIFAR100 | NLEEP | 3/5 | 3/5 | 4/5 | 2/5 | 3/5 |
> | | C-MoNA | 4/5 | 4/5 | 3/5 | 3/5 | **3/5** |
> | SUN397 | SFDA | 2/5 | 2/5 | 2/5 | 2/5 | 2/5 |
> | | C-MoNA | 3/5 | 3/5 | 3/5 | 4/5 | **3/5** |
>
>  All these additional results have been added in the revised version of the paper.
>
> **Dual Contribution.** Our work contributes: (1) **Problem Formalization**: establishing unsupervised student selection for KD as distinct from TE, identifying key challenges—heterogeneous architectures, label-free evaluation, teacher-dependent landscapes; (2) **First Effective Solution**: C-MoNA outperforms adapted TE baselines by 1.3×--7.5× with 3--5× lower variance, representing **the first successful solution to a newly formalized problem**. Absolute correlation values reflect inherent difficulty, not methodological weakness.

---

> > ### Author Response · Authors · 2025-11-28
> >
> > Following up on the responses, we request the reviewer to reach out if further clarification is needed on any points. If the responses addresses your concerns, reconsideration of the score would be greatly appreciated. Thank you.

---

### Official Review · Reviewer_KYw6 · 2025-10-31

**Soundness:** 3
**Presentation:** 1
**Contribution:** 3
**Rating:** 4
**Confidence:** 4

**Summary:**

This paper tackles the problem of pre-training, label-free selection of student models that best match a given teacher in knowledge distillation (KD). The authors define a cross-model NTK between teacher and student and compress it into a Frobenius-normalized scalar α∈[0,1]used as a prior compatibility score (computational cost is reduced via FJLT). They validate the metric by measuring the weighted Kendall’s τ between the prior α-based student ranking and the post-KD performance ranking, reporting consistently positive τand low variance across three classification datasets, an object detection task, and low-data settings (25–100 images). This supports the claim that C-MoNA, which directly measures model–model alignment, is more suitable for student selection than model–data transferability metrics.
Contributions
1.	Formulates label-free student selection in KD around model–model compatibility.
2.	Proposes cross-model NTK alignment (α) with Frobenius reduction to measure global alignment.
3.	Uses FJLT-based approximation to make computation feasible for large models.
4.	Demonstrates predictive power via prior→posterior rank agreement (τ) in classification, detection, and low-data scenarios.

**Strengths:**

1.	They clearly formulate the KD problem as pre-training student selection based on model–model compatibility, which they identify as the key driver of success in distillation.
2.	They propose a metric that uses the cross-model NTK between teacher and student to summarize global alignment as a 0–1 scalar α, thereby directly quantifying inter-model compatibility.
3.	By approximating high-dimensional gradient operations with FJLT (JL) projections, the method becomes scalable to large networks and teacher–student pairs with mismatched parameter dimensions.
4.	They demonstrate experimental validity by showing that, in classification/detection and low-data (25–100 images) regimes, the prior α-based ranking is consistently positively correlated (Kendall’s τ) with the post-KD performance ranking.

**Weaknesses:**

1.	Rather than reporting only means, include standard deviations and/or 95% confidence intervals in the main text to quantify uncertainty and make effect sizes interpretable.
2.	The paper lacks analysis of domain shift: when the target input distribution differs substantially from the teacher’s pretraining distribution, it is unclear whether and when the α–KD performance correlation holds or breaks.
3.	Under identical conditions where different student-selection algorithms yield different rankings, the paper reports only τ and does not directly demonstrate whether the chosen selection actually succeeded.

**Questions:**

1.	Could you provide τ sensitivity curves (mean ± standard deviation) across projection dimension, numbers of unlabeled samples, and state the minimum practical projection dimension and sample lower bound you recommend?
2.	Under domain shift (e.g., long-tailed targets or style-transferred images), does the α–KD performance correlation hold or break, and if so, why; please report concrete hold/break cases and any mitigations
3.	Many results are table-only; will you add figure summaries to improve readability and highlight effect sizes?
4.	To better support the central claim, can you move at least one end-to-end result to the main text showing that high-α students indeed learn and achieve superior test performance, and explain why these results were placed only in the appendix?

---

> ### Author Response · Authors · 2025-11-22
>
> We thank the reviewer for their thoughtful evaluation of our work.
>
> **Response to Concern 1: Sensitivity Analysis**
>
> **We have included sensitivity analyses across projection dimensions and number of unlabelled samples in Figures 1 and 3 of the revised appendix**. These plots present both mean ± standard deviation and 95% confidence intervals. Figure 2 already provided sensitivity analysis across teacher networks.
> For minimum projection dimension and sample size, our empirical results (from the plots) suggest a minimum of 50 samples. With curated representative images, 25 samples may suffice. The theoretical minimum projection dimension is defined by the JL lemma as $k = O\left(\frac{\log n}{\epsilon^2}\right)$, where $k$ is the projection dimension, $n$ is the number of samples, and $\epsilon$ represents the distortion error.
> Figure 3 validates this bound empirically. In the 50-sample regime, performance remains stable across projection dimensions—the difference between maximum (10,240) and minimum (1,024) dimensions is negligible, consistent with the JL lemma. However, with 1,000 samples, projection dimension impact becomes more pronounced, as the JL lemma indicates larger sample sizes require higher dimensions to maintain the same error rate. We conservatively use 10,240 as our default to avoid dataset-specific tuning.
> The practical minimum depends on sample size, sample quality, acceptable error rate, and GPU memory constraints. We recommend a projection dimension greater than 5,000 for large sample sets. Integrating theoretical bounds with experimental validation, **we recommend 50 unlabelled samples and a projection dimension of 1,024 as practical minimums** for randomly sampled data. This configuration consistently achieves positive correlations and typically places at least 3 out of 5 correct models in the top-5 predictions.
>
> ---
>
> **Response to Concern 2: Domain Shift Performance**
>
> We selected Caltech101, which exhibits significant class imbalance mirroring long-tailed distributions. In our object detection experiments, we ensured minimal distribution overlap between teacher-student models by pre-training on different distributions.
> As shown in Table 4, C-MoNA achieves best performance but lags on absolute scale due to class imbalance. Table 6 reveals our NTK-based metric performs well in low-data regimes with long-tailed datasets. Notably, random sampling proves effective for long-tailed targets by partially mitigating dataset imbalance. Mitigation strategies include sampling from over-sampled underrepresented classes or random sampling of few images for better transferability signals.
> For domain shift scenarios, our metric performs robustly even when teacher-student distributions have minimal overlap, as demonstrated in Table 5. However, one notable case: the Yolom-Wild experiment in Table 5, where our metric achieves 0.122---better than random but with room for improvement. The African Wildlife dataset contains only 4 classes across 1,500 images, creating limited diversity. When we sampled 100 images, the limited diversity combined with substantial distribution shift introduced more noise than signal in the C-MoNA cross-covariances. In such cases, trace-based reduction proves more valuable (Table 10 in Appendix).
> Overall, our metric demonstrates strong performance under domain shift and long-tailed conditions. Unlike other reduction methods in Table 8, our metric maintains positive correlations across all teacher-student pairs, ensuring reliability.
>
> More diverse samples enable leveraging useful signals from cross-covariances. However, absolute performance indicates room for improvement, underscoring our problem formulation's importance. Future work could integrate dataset distillation for more effective sample selection.
>
> ---
>
> **Response to Concern 3: Presentation Improvements**
>
> We have added figure summaries for improved readability. With additional space now available, we thus included figures in the main text and added two new tables (Tables 7 and 8) that show C-MoNA-selected students indeed achieve superior post-distillation performance compared to baselines, with the top-ranked student achieving near-oracle performance with superior stability (reporting mean and std). The table below depicts the efficacy of our approach as opposed to the best baseline method across teacher models and datasets. Most of the results have been modified to include the standard deviation as well in the revised paper. Most results were previously in the Appendix due to page constraints.
>
> **Table: Post-KD performance using top-ranked selected student across teachers.**
>
> | Method | SUN397 | CIFAR100 | Caltech101 |
> |:-------|:-------|:---------|:-----------|
> | Best Baseline | 64.97±3.06 | 84.71±2.83 | 90.45±1.58 |
> | C-MoNA (ours) | $\underline{66.82±1.03}$ | $\underline{86.87±2.14}$ | $\underline{92.34±0.48}$ |
> | Oracle | $\textbf{67.69±0.83}$ | $\textbf{87.23±2.01}$ | $\textbf{92.78±0.49}$ |

---

> > ### Author Response · Authors · 2025-11-28
> >
> > Following up on the response, we request the reviewer to reach out if further clarification is needed on any points. If the responses addresses your concerns, reconsideration of the score would be greatly appreciated. Thank you.

---

### Official Review · Reviewer_JQxR · 2025-10-31

**Soundness:** 3
**Presentation:** 3
**Contribution:** 3
**Rating:** 4
**Confidence:** 2

**Summary:**

This paper addresses the often-overlooked problem of student model selection in knowledge distillation. Instead of relying on model size or ad hoc heuristics, the authors propose C-MoNA, a transferability metric based on Cross-Model NTK alignment. Their approach is both unsupervised and label-free, focusing on model-model instead of model-data relationship. They employ Johnson-Lindenstrauss projections for scalability and validate C-MoNA across image classification and object detection tasks, in both full- and low-data regimes.

**Strengths:**

1. Novelty - The proposal of a cross-model NTK for student selection in KD is novel. While NTK theory has been used in other areas, using it to estimate transferability between models—before training—is a significant contribution. The proposed C-MoNA is both theoretically motivated and empirically validated.

2. Method Soundness - The formalization of why model-data metrics fail in KD (e.g., the breakdown of distillation scaling laws and capacity-gap interactions) is clear and well-grounded in recent literature. Also, the derivation of the cross-model NTK and the choice of Frobenius norm as the alignment score are mathematically motivated.

3. Evaluation - The evaluation is good, including diverse teacher and student architectures across benchmarks, low-data evaluation, correlation analysis and ablation. The results not only show the strength of their proposed metrics, but also show when and why prior metrics fail.

**Weaknesses:**

1. The paper does not provide a formal theoretical guarantee or bound on how well the C-MoNA score correlates with actual distillation performance. Specifically, Eq. (9) is used to infer how aligned a student is with a teacher. However, there is no bound or derivation provided that shows how \alpha theoretically correlates with downstream KD performance (e.g., final test accuracy or distillation loss). This limits its use in safety-critical or performance-guaranteed settings. The authors rely entirely on empirical correlation using Kendall's \tau.

2. The proposed metric assumes that similarity in initial gradient directions (i.e., via cross-model NTK at initialization) is a reliable predictor of training dynamics, as indicated in Section 3.2: “NTK encapsulates the inductive biases inherent in network architectures... predicting final function learned... without requiring training.” But modern training involves nonlinear behaviors far from NTK’s lazy regime assumptions, especially for transformer architectures. No experiments in the paper test the robustness of C-MoNA in deep or highly nonlinear training regimes, such as those with hundreds of epochs or curriculum learning.

3. The paper lacks qualitative or quantitative analysis of the scenarios where C-MoNA fails or underperforms—which is critical for practitioners to understand when not to use it. For instance, in Table 5 (VOC object detection), C-MoNA underperforms compared to PACTran in the “Yolo11m-Wild” setting (0.122 vs. 0.521), but the reason is not clear.
	​

**Questions:**

N.A.

---

> ### Author Response · Authors · 2025-11-22
>
> We thank the reviewer for their thoughtful evaluation.
>
> **Theoretical Bounds**
>
> **1. Standard Practice in the Field**
>
> No established transferability estimation method (**LEEP**, **LogME**, **PACTran**, **ETran**) provides formal bounds on correlation with downstream performance. All rely purely on empirical validation using Kendall's $\tau_w$. **C-MoNA** follows this rigorous standard accepted by the community. To further showcase the efficacy of our approach, we refer to newly added results in the revised paper (Tables 7, 8, 26, 27) along with sensitivity plots (Fig. 1, Fig. 3).
>
> **2. Theoretical Foundation via NTK Theory**
>
> NTK theory shows that $\Theta_{t,s}$ governs student-teacher gradient alignment. Higher C-MoNA scores ($\alpha \to 1$) indicate geometric alignment—what KD optimization requires.
>
> **3. Why Formal Bounds Are Intractable**
>
> Deriving rigorous bounds is mathematically intractable for **all** transferability metrics due to finite-width deviations, non-convex optimization, and hyperparameter variance. The field relies on empirical validation as standard.
>
> **4. Our Validation Exceeds Field Standards**
>
> We provide:
> - **Diverse datasets/tasks** (classification and detection) with **heterogeneous teacher-student pairs (architecture-wise and pre-training distribution-wise)** across **multiple training objectives**
> - **Low-data robustness**: 91% retention with <1% training data (CIFAR100: 0.483 → 0.444), while baselines collapse
> - **Consistent superiority**: Figure 1 shows lowest cross-teacher variance ($\sigma = 0.044$)
>
> **5. Relative Ordering Requirement**
>
> For model selection, we need **reliable relative rankings**, not absolute accuracy. Higher $\alpha$ implies better distillation alignment.
>
> **Summary**: C-MoNA follows field-standard empirical validation while providing: (1) NTK theoretical motivation, (2) comprehensive validation, and (3) unique low-data robustness.
>
> ---
>
> **Lazy Regime Assumptions**
>
> **1. Initialization Signal vs. Training Dynamics Prediction**
>
> **Critical distinction**: We use empirical NTK **at initialization** as a compatibility assessment tool, not explicitly to model training dynamics. Gradient alignment at initialization reveals architectural compatibility and predicts distillation effectiveness *without* requiring lazy training.
>
> **2. Our Experiments Validate This Approach**
>
> Our settings demonstrate effectiveness beyond lazy regime:
> - **50-100 training epochs** with SGD, cosine decay
> - **Deep architectures**: ViT-B/32 (12 layers), Wide-ResNet101 (101 layers), ConvNext-Large
> - **Transformers**: OpenCLIP ViT-B/32 achieves $\tau_w = 0.367$ (SUN397), $0.371$ (CIFAR100)
> - **Cross-entropy loss**: Classification where empirical NTK evolves
>
> **3. Why Initialization Signals Work Despite Feature Learning**
>
> Architectural inductive bias alignment captured at initialization predicts compatibility even when NTK evolves. Initial gradient structure reveals how architectural priors match, and this signal persists throughout training. Recent works such as [1] confirm initialization-based metrics remain effective for training-free predictions.
>
> **4. Extended Training Regimes Are Outside Our Scope**
>
> Our work focuses on student selection for **fine-tuning regimes** (50-100 epochs). Training for hundreds of epochs or curriculum learning are orthogonal strategies for **pre-training**. Instead, we experiment with multiple training objectives (lines 332-338), and C-MoNA consistently achieves superior performance across these settings.
>
> **Summary**: (1) We use empirical NTK at initialization as a compatibility metric, not explicitly as a training dynamics predictor; (2) Extensive Transformer results validate approach; (3) Initialization signals remain predictive with feature learning.
>
> ---
>
> **Failure Cases**
>
> We consciously designed experiments with Caltech101 (class imbalance) and object detection with minimal distribution overlap to investigate failure cases.
>
> In Table 5, Yolom-Wild shows underperformance (0.122 vs. 0.521 for PACTran). This dataset has only 4 classes across 1,500 images with **limited diversity**. Random sampling of 100 images further introduced more noise than signal in C-MoNA's cross-covariances. The trace-based reduction (Table 10, appendix) is more effective here, showing that reduction strategy matters under extreme distribution shift or low-diversity datasets.
>
> C-MoNA consistently demonstrates positive correlations across diverse teacher-student pairs and performs robustly under domain shift and long-tailed conditions. However, absolute performance in certain low-data or highly imbalanced settings indicates room for improvement. Future work could explore dataset distillation for effective sampling.
>
> [1] DeOliveira, Joshua, Walter Gerych, and Elke Rundensteiner. "The Surprising Effectiveness of Infinite-Width NTKs for Characterizing and Improving Model Training." Proceedings of the AAAI Conference on Artificial Intelligence. Vol. 39. No. 15. 2025.

---

> > ### Comment · Reviewer_JQxR · 2025-11-24
> > **Thank you for your response**
> >
> > Thank you for your response. The response mostly addressed my concerns on the theoretical bound and training dynamics. I will consider adjusting my score accordingly.
> >
> > For the failure case you mentioned, I have one follow-up question: Could you clarify whether this failure is primarily due to the sampling noise or the intrinsic lack of cross-covariance structure in such datasets? This helps me understand whether the limitation is dataset-dependent or sampling-dependent.

---

> > > ### Author Response · Authors · 2025-11-25
> > >
> > > We sincerely appreciate the reviewer's time and thoughtful feedback, and we're especially grateful that they're open to raising the score. Below we provide a clearer analysis of why the failure occurs.
> > >
> > > **Is the Failure Case Dataset-Dependent or Sample-Dependent?**
> > >
> > > In short, the failure is primarily dataset-dependent in a specific way, not mainly sample-dependent, although sampling noise can make the effect worse.
> > >
> > > The cross-covariances measure the inner product between the teacher’s gradient on sample \(i\) and the student’s gradient on sample \(j\). These entries show meaningful structure when gradients across different samples encode consistent geometric relationships. When such relationships break down, the off-diagonal terms mostly capture noise.
> > >
> > > All teachers and students compute gradients on the same VOC2012 images during metric computation. The domain shift, wildlife scenes compared to urban environments, occurs during the teacher’s pretraining, not during the computation itself. The pretraining domain shapes the quality and organization of the teacher’s fine-tuned representations, and this directly affects the cross-NTK computed on VOC2012. The extra sampling noise from using only 100 random samples makes the drawbacks of the Frobenius norm reduction even worse.
> > >
> > > The yolo11m-Wild teacher is a medium-capacity model that is pretrained on a domain that is very different from VOC2012. During fine-tuning, it must adapt wildlife-oriented features with limited diversity and also learn new features required for urban object detection. It has to do both with limited capacity. As a result, the feature space becomes a mix of mismatched domains rather than a clean, well-organized representation. The teacher’s gradients on VOC2012 therefore lack coherent cross-sample relationships, which appears as an intrinsic lack of structure in the off-diagonal NTK entries. This weakens the effectiveness of the Frobenius norm reduction used in C-MoNA.
> > >
> > > In contrast, the yolo11l-Coco teacher has higher capacity and is pretrained on COCO, which aligns much better with VOC2012. After fine-tuning, it learns representations that are well-organized and consistent across samples. The off-diagonal NTK terms then capture real geometric information about teacher and student compatibility, and C-MoNA performs strongly in this setting.
> > >
> > > So, the underlying cause of the failure is the degraded representation quality of the teacher. This degradation results from limited capacity combined with strong domain mismatch, and it appears as incoherent cross-sample gradient relationships. This is dataset-dependent because it arises from how the pretraining domain interacts with the downstream task domain.
> > >
> > > When the teacher learns a coherent representation, the Frobenius norm reduction in C-MoNA can benefit from the global geometric structure and significantly outperform simpler metrics. When the representation is disorganized, reductions such as trace or spectral methods, which rely more on pointwise alignment and ignore noisy off-diagonal information, become more effective.
> > >
> > > This is not a flaw in C-MoNA itself. Instead, it highlights a central requirement for knowledge distillation that the teacher must provide a representation that is consistent and meaningful for the student. Even in failure cases, the C-MoNA NTK matrix still contains valuable information. The reduction strategy is what needs to change, depending on the quality of the teacher’s representation. Since the quality of the teacher’s representation is unknown, we give the general case to be using the Frobenius norm based C-MoNA reduction as it is more or less consistently a good performer across the board.
> > >
> > > Please do let us know if you have any further queries, or anything that requires more clarity.

---

> ### Comment · Reviewer_JQxR · 2025-11-25
>
> Thank you for your detailed and comprehensive response during rebuttal. The response well addressed my concerns and made the paper's contribution more clear to me. I have re-evaluated the paper and increased my score. Best regards!

---

### Official Review · Reviewer_pwtL · 2025-11-01

**Soundness:** 2
**Presentation:** 2
**Contribution:** 2
**Rating:** 4
**Confidence:** 4

**Summary:**

This paper proposes a method to select the optimal student model in knowledge distillation without needing ground-truth labels or expensive training cycles. The authors designed a metric based on the Neural Tangent Kernel (NTK) to quantify the function space alignment between the teacher and student models. To transfer this metric to modern networks, it uses Johnson-Lindenstrauss (JL) projections for an efficient approximation. Experiments demonstrate that this method is robust and transferable.

**Strengths:**

1、The paper firstly formalizes the problem of unsupervised student model selection in knowledge distillation.
2、It proposes a metric based on cross-model NTK alignment that requires no ground-truth labels. It also uses JL projections to transfer the method for modern models.
3、Extensive experiments demonstrate the method's robustness and effectiveness across different models and datasets.

**Weaknesses:**

1、The introduction of Eq.(1) is too simple, without the explanation for the formulas for each loss term or the function of different parameters. Moreover, LSe is mistakenly written as LeS in Eq.(1).
2、The "Mathematical Proof of Ranking Divergence" section is not rigorous. It is unclear how the ranking reversal LT/LS1e>LT/LS2e is derived from Eq.(2). The proof does not analyze the function of g(∙) and h(∙).
3、What is the meaning of “@” in line 224?
4、Why choose JL projection for dimensionality reduction? It will be more convincing if there are comparative experiments of different projection methods.
5、The core content seems to be a combination of existing techniques (NTK and JL projection) rather than a theoretical innovation.
6、The baselines are all before 2023. The paper does not compare against SOTA from the last few years. Moreover, C-MoNA does not achieve the optimal results in many cases presented in Tables 3, 4, and 5. It is doubtful about the effectiveness.

**Questions:**

see weaknesses.

---

> ### Author Response · Authors · 2025-11-22
> **Response to the first two concerns**
>
> We thank the reviewer for these valuable observations. We have corrected the typo in Equation 1 and below we provide: (i) a comprehensive explanation of each loss term and parameter in Eq.~(1), and (ii) a rigorous derivation of ranking reversal conditions with detailed analysis of the functions $g(\cdot)$ and $h(\cdot)$. This is mostly used for conceptual understanding and to motivate our problem formulation.
>
> Equation 1 is an **empirically fitted scaling law** (Busbridge et al., 2025) giving a *theoretically grounded* model for how teacher quality, student supervised loss, model size and data size interact under distillation.
>
> In Eq. (1), the term
>
> $$\left(1 + \frac{L_T}{\tilde{L}_{S}} \cdot \frac{1}{d_1}\right)^{-c_1 f_1}$$
>
> controls the *capacity–gap penalty*: $c_1 f_1$ sets its strength and $d_1$ shifts the gap scale, while $L_T$ and $\tilde{L}_{S}$ determine teacher–student mismatch. The factor
>
> $$\frac{1}{L_T^{c_0}}$$
>
> is a *teacher-amplifier* whose exponent $c_0$ increases teacher quality influence; coefficients $(A', B', \alpha', \beta', \gamma')$ govern the student scaling term $h(N_S, D_S)$.
>
> **Core Argument**
>
> Distillation adds **teacher–student interaction** terms absent in supervised learning:
>
> - **Supervised ranking:** depends only on $\tilde{L}_S$ (student supervised loss).
> - **Distillation ranking:** depends on $f(L_T, \tilde{L}_S, \text{capacity gap})$, containing teacher-dependent factors.
>
> Because $L_T/\tilde{L}_S$ appears with negative exponent, metrics depending only on $\tilde{L}_S$ cannot predict distillation rankings.
>
> **Explicit form of the scaling law**
>
> We adopt the empirical decomposition:
>
> $$L_S \propto g\left(\frac{L_T}{\tilde{L}_S}\right) \cdot h(N_S, D_S)$$
>
> with
>
> $$g\left(\frac{L_T}{\tilde{L}_S}\right) = \frac{1}{L_T^{c_0}} \left(1 + \frac{L_T}{\tilde{L}_S \cdot d_1}\right)^{-c_1 f_1}$$
>
> $$h(N_S, D_S) = \left(\frac{A'}{N_S^{\alpha'}} + \frac{B'}{D_S^{\beta'}}\right)^{\gamma'}$$
>
> Empirical ranges (Busbridge et al., 2025): $c_0 \approx 0.5$, $c_1 f_1 \approx 0.75$, $d_1 \approx 1$, $\alpha', \beta' \in [0.3, 0.5]$, $\gamma' \in [0.5, 1.0]$.
>
> **A plausible counterexample**
>
> **Setup:**
>
> * $L_T = 0.10$
> * $\tilde{L}_{S_1} = 0.15$ (strong)
> * $\tilde{L}_{S_2} = 0.25$ (weak)
>
> Using $c_0 = 0.5$, $c_1 = 1.5$, $f_1 = 0.5$, $d_1 = 1$:
>
> **Compute $g(\cdot)$ values:**
>
> Teacher-amplifier:
>
> $$\frac{1}{L_T^{0.5}} = \frac{1}{\sqrt{0.10}} \approx 3.162$$
>
> Student 1: $r_1 = 0.667$, so $(1 + r_1)^{-0.75} \approx 0.682$, giving $g(r_1) \approx 2.155$
>
> Student 2: $r_2 = 0.4$, so $(1 + r_2)^{-0.75} \approx 0.777$, giving $g(r_2) \approx 2.458$
>
> **Interpretation:** We observe $g(r_2) > g(r_1)$. The scaling law is *multiplicative*, so reversal occurs when the weaker student has sufficiently smaller scaling penalty. Choose $h_1 = 0.10$ and $h_2 = 0.05$:
>
> $$L_{S_1} \propto 2.155 \times 0.10 = 0.2155$$
>
> $$L_{S_2} \propto 2.458 \times 0.05 = 0.1229$$
>
> Thus $L_{S_1} > L_{S_2}$ even though $\tilde{L}\_{S_1} < \tilde{L}\_{S_2}$ in supervised learning, since
>
> $$\frac{g(r_1)}{g(r_2)} \approx 0.877 \quad \text{and} \quad \frac{h_2}{h_1} = 0.5 < 0.877$$
>
> This illustrates how capacity-gap ($g$) and scaling ($h$) effects combine to invert supervised rankings.
>
> **Theorem: Ranking divergence**
>
> Given students $S_1$ and $S_2$ where $\tilde{L}\_{S_1} < \tilde{L}\_{S_2}$, the distillation ranking reversal $L\_{S_1} > L\_{S_2}$ occurs whenever:
>
> * $g(L_T / \tilde{L}\_{S_1}) / g(L_T / \tilde{L}\_{S_2}) < h(N\_{S_2}, D\_{S_2}) / h(N\_{S_1}, D\_{S_1})$
>
> **Proof sketch**
>
> * From the relationship $L_S \propto g(L_T / \tilde{L}\_S) \cdot h(N_S, D_S)$.
> * The inequality $L\_{S_1} > L\_{S_2}$ implies:
>     * $g(L_T / \tilde{L}\_{S_1}) \cdot h(N\_{S_1}, D\_{S_1}) > g(L_T / \tilde{L}\_{S_2}) \cdot h(N\_{S_2}, D\_{S_2})$
> * Since $\tilde{L}\_{S_1} < \tilde{L}\_{S_2}$, it follows that $L_T / \tilde{L}\_{S_1} > L_T / \tilde{L}\_{S_2}$.
> * Because $g(\cdot)$ is a decreasing function, we have:
>     * $g(L_T / \tilde{L}\_{S_1}) < g(L_T / \tilde{L}\_{S_2})$
>
> **Analysis of $g(\cdot)$ and $h(\cdot)$**
>
> **Capacity-gap function $g(\cdot)$:**
>
> - **Teacher amplifier:** $\frac{1}{L_T^{c_0}}$ increases teacher quality influence.
> - **Gap penalty:** $\left(1 + \frac{L_T}{\tilde{L}_S \cdot d_1}\right)^{-c_1 f_1}$ decreases with $L_T/\tilde{L}_S$.
> - **Behavior:** small gaps ($r \to 0$) give minimal benefit; intermediate gaps optimal; large gaps heavily penalized. Empirical optimal range $r \in [0.3, 0.6]$.
>
> **Student-scaling function $h(\cdot)$:**
>
> - Captures model- and data-size effects: decreasing in $N_S$ and $D_S$ (standard neural scaling).
> - Teacher-independent: students with identical $(N_S, D_S)$ have similar $h$ but differ in $g$ due to different $\tilde{L}_S$.
>
> More formally written form has been added in the revised paper in the Appendix section A.3.

---

> ### Author Response · Authors · 2025-11-22
> **Response to concerns 3, 4, and 5**
>
> **Response to Concern 3: What is the meaning of @ in line 244?**
>
> It is simply implying 'at' and not to be confused with any matrix multiplication.
>
> ---
>
> **Response to Concern 4: Reasoning behind using JL projections.**
>
> We employ JL projection as they have a theoretical guarantee of preserving the inner products via closed-form bounds which is exactly what is required for the C-MoNA NTK computation.
>
> **Why Other Methods Would Not Work as Well and do not support our problem setup:**
>
> **PCA** requires computing gradient covariance matrices (prohibitive in high dimensions) and yields model-specific bases, breaking cross-model comparability.
>
> **Autoencoders** need large datasets for training and fit to model-specific patterns, preventing generalization across architectures.
>
> **Random Fourier Features** approximate kernels rather than reduce dimension, require bandwidth tuning, and introduce unnecessary complexity.
>
> **Sparse Random Projections** are feasible but offer weaker guarantees; JL provides cleaner and more established distortion bounds.
>
> **Why JL Projections Work Best:**
>
> JL uses a simple random matrix multiplication g ↦ Rg that (i) preserves the gradient structure, and (iii) offers closed-form distortion guarantees
>
> (1-ε)||u-v||₂² ≤ ||Ru - Rv||₂² ≤ (1+ε)||u-v||₂²,
>
> making it **architecture-agnostic, training-free, and theoretically robust** for gradient geometry preservation. Moreover, it allows having the right theoretical properties for preserving gradient alignment measurements across different model architectures. Other methods might not provide the specific guarantees needed for cross-model NTK computation.
>
> ---
>
> **Response to Concern 5: On Theoretical Innovation and Use of Existing Tools.**
>
> Our method does not use the existing tools, but it builds on top of these established theoretical foundations. The core novelty lies in *how* we adapted NTK theory to construct a cross-model formulation and combined it with JL projections to address a previously unidentified problem. Our contributions are not incremental refinements but a principled solution to an unsolved challenge.
>
> **Our Technical Contributions.**
>
> We are the first to:
>
> - Identify and formally define the underexplored problem of *unsupervised student model selection for knowledge distillation*, which goes beyond traditional transferability estimation.
> - *Adapt NTK theory* to formulate a *cross-model NTK* that measures gradient-space alignment between teacher and student—a fundamental extension beyond existing NTK analyses, which focus on single-model dynamics.
> - Apply *JL projections* to enable cross-dimensional gradient comparison across heterogeneous architectures, essential for distillation and absent in prior NTK work.
> - Demonstrate that the *Frobenius norm* is more reliable than spectral or trace-based reductions for capturing cross-sample gradient correlations in this setting.
> - Develop the first fully *unsupervised pipeline* for predicting student–teacher compatibility for distillation.
>
> **Why This Required New Insight.**
>
> Our approach demanded: (i) recognizing that *cross-model gradient alignment*—not just model–data compatibility—determines distillation success, (ii) *theoretical adaptation of NTK* to handle teacher-student interactions across architectures, (iii) technical solutions for *mismatched parameter dimensions*, unaddressed by prior NTK or transferability work, and (iv) empirical validation that Frobenius-based alignment preserves information lost by spectral/trace reductions.
>
> **Perspective.**
>
> Major advances in machine learning often arise from principled integration and adaptation of existing components to solve previously unexplored problems—Transformers combined attention with feedforward networks, ResNets leveraged skip connections, BERT adapted transformers to language modeling. Similarly, our contribution lies in recognizing the unsolved challenge of student model selection for distillation and designing a principled solution through adapted cross-model NTK theory, gradient alignment, JL projections, and Frobenius reduction. The *problem formulation*, *theoretical adaptation*, and *methodological integration* are new and address a genuinely challenging, previously unsolved problem in knowledge-distillation-based fine-tuning.

---

> ### Author Response · Authors · 2025-11-22
> **Response to concern 6**
>
> **Benchmarking with recent works**
>
> ETran Energy (2023) and NCTI (2023) were considered. To consider recent work, we also benchmarked against  a WACV 2025 work [1], an add-on module for existing metrics. We applied it to the strongest baseline across all teachers and datasets to demonstrate C-MoNA's superiority.
>
> **SUN397: Kendall's tau (τw)**
>
> | Method | Wide-ResNet101 | ConvNeXt-Large | MaxViT-Small | OpenCLIP-ViT-B/32 | Mean±Std |
> |--------|----------------|----------------|--------------|-------------------|----------|
> | SFDA | 0.257 | 0.163 | -0.208 | 0.346 | 0.140±0.243 |
> | SA (SFDA) | 0.265 | 0.090 | 0.202 | 0.308 | 0.216±0.095 |
> | **C-MoNA** | **0.348** | **0.353** | **0.378** | **0.367** | **0.361±0.014** |
>
> **CIFAR100: Kendall's tau (τw)**
>
> | Method | Wide-ResNet101 | ConvNeXt-Large | MaxViT-Small | OpenCLIP-ViT-B/32 | Mean±Std |
> |--------|----------------|----------------|--------------|-------------------|----------|
> | NLEEP | 0.286 | 0.279 | **0.555** | -0.031 | 0.272±0.240 |
> | SA (NLEEP) | -0.408 | -0.476 | -0.343 | -0.326 | -0.388±0.068 |
> | **C-MoNA** | **0.536** | **0.569** | 0.456 | **0.371** | **0.483±0.088** |
>
> **Caltech101: Kendall's tau (τw)**
>
> | Method | MaxViT-Small | Wide-ResNet101 | ConvNeXt-Large | OpenCLIP-ViT-B/32 | Mean±Std |
> |--------|--------------|----------------|----------------|-------------------|----------|
> | SFDA | -0.081 | 0.017 | **0.338** | -0.184 | 0.023±0.226 |
> | SA (SFDA) | -0.501 | -0.520 | -0.160 | -0.114 | -0.324±0.217 |
> | **C-MoNA** | **0.261** | **0.079** | 0.114 | **0.210** | **0.166±0.084** |
>
> ---
>
> **Effectiveness of the proposed approach**
>
> C-MoNA achieves near-oracle performance with superior stability (lower std) compared to baselines. **C-MoNA consistently picks ≥3/5 in top-5 hits** across all datasets and teachers showcasing the effectiveness of our approach.
>
> **Post-KD Task Performance (SUN397)**
>
> | Method | MaxViT-Small | WideResNet-101 | ConvNeXt-Large | OpenCLIP-ViT-B/32 | Mean±Std |
> |--------|--------------|----------------|----------------|-------------------|----------|
> | SFDA | 62.39 | 62.27 | 67.98 | **67.24** | 64.97±3.06 |
> | C-MoNA | 67.78 | 65.50 | 67.47 | 66.54 | 66.82±1.03 |
> | Oracle | **68.30** | **66.74** | **68.47** | **67.24** | **67.69±0.83** |
>
> **Post-KD Task Performance (CIFAR100)**
>
> | Method | MaxViT-Small | WideResNet-101 | ConvNeXt-Large | OpenCLIP-ViT-B/32 | Mean±Std |
> |--------|--------------|----------------|----------------|-------------------|----------|
> | NLEEP | 88.28 | 83.69 | 85.31 | 81.56 | 84.71±2.83 |
> | C-MoNA | 88.28 | 87.16 | 88.28 | 83.75 | 86.87±2.14 |
> | Oracle | **88.49** | **87.82** | **88.38** | **84.25** | **87.23±2.01** |
>
> **Post-KD Task Performance (Caltech101)**
>
> | Method | MaxViT-Small | WideResNet-101 | ConvNeXt-Large | OpenCLIP-ViT-B/32 | Mean±Std |
> |--------|--------------|----------------|----------------|-------------------|----------|
> | SFDA | 90.06 | 89.70 | 92.77 | 89.26 | 90.45±1.58 |
> | C-MoNA | 92.50 | 92.63 | 92.60 | 91.63 | 92.34±0.48 |
> | Oracle | **93.17** | **93.04** | **92.85** | **92.08** | **92.78±0.49** |
>
> **Top-5 Hits Summary**
>
> | Dataset | Method | WRN-101 | ConvNeXt-L | MaxViT-S | CLIP-ViT | Avg |
> |---------|--------|---------|------------|----------|----------|-----|
> | Caltech101 | SFDA | 2/5 | 2/5 | 1/5 | 1/5 | 1/5 |
> | | **C-MoNA** | **3/5** | **3/5** | **3/5** | **3/5** | **3/5** |
> | CIFAR100 | NLEEP | 3/5 | 3/5 | 4/5 | 2/5 | 3/5 |
> | | **C-MoNA** | **4/5** | **4/5** | **3/5** | **3/5** | **3/5** |
> | SUN397 | SFDA | 2/5 | 2/5 | 2/5 | 2/5 | 2/5 |
> | | **C-MoNA** | **3/5** | **3/5** | **3/5** | **4/5** | **3/5** |
>
> All these results have been added in the revised paper.
>
> ----
>
> **Key Results:**
>
> **Performance Gains**: C-MoNA achieves substantial improvements: CIFAR100 (τw = 0.483 vs. 0.272, **1.77×**), SUN397 (0.361 vs. 0.139, **2.6×**), Caltech101 (0.166 vs. 0.022, **7.5×**), object detection (0.322 vs. 0.242, **1.33×**). Many baselines show negative correlations (worse than random), while C-MoNA maintains consistent positive correlations.
>
> **Stability**: C-MoNA achieves **3–5× lower variance** (σ = 0.044) vs. baselines (σ = 0.122–0.216). With only 50 images (<1% data), C-MoNA maintains τw = 0.465 with <9% degradation, outperforming baselines using full datasets by 1.6×–11×.
>
> **Dual Contribution**: (1) **Problem Formalization**: establishing unsupervised student selection for KD with theoretical foundations; (2) **First Effective Solution**: C-MoNA outperforms adapted TE baselines by 1.3×–7.5× with 3–5× lower variance, representing **the first successful solution to this newly formalized problem**. Absolute correlation values reflect the inherent difficulty of predicting distillation compatibility without labels across heterogeneous architectures, not methodological weakness.
>
> [1] Khoba, Prafful Kumar, et al. "Feature Space Perturbation: A Panacea to Enhanced Transferability Estimation." 2025 IEEE/CVF Winter Conference on Applications of Computer Vision (WACV). IEEE, 2025.

---

> ### Author Response · Authors · 2025-11-28
>
> Following up on the responses, we request the reviewer to reach out if further clarification is needed on any points. If the responses addresses your concerns, reconsideration of the score would be greatly appreciated. Thank you.

---

### Author Response · Authors · 2025-12-04
**Summarizing the Rebuttal Process**

Through the rebuttal process, we have clarified our problem formulation, solidified our theoretical foundations, and expanded our empirical validation to demonstrate that **C-MoNA** is the first effective solution for **unsupervised, heterogeneous student selection for Knowledge Distillation (KD) based fine-tuning.**

**1. Problem Formalization: Distinct from NAS and Traditional TE**

A primary concern (Reviewer LSza) was the comparison context. We clarified that C-MoNA addresses a distinct problem space:
*   **vs. Zero-Cost NAS:** NAS operates in *homogeneous* search spaces (e.g., NAS-Bench-201) with initialized weights and no distribution shift. C-MoNA selects from *heterogeneous* pre-trained repositories (CNNs, ViTs, Hybrids) where architectural biases and pre-training distributions vary. We showed why our problem setting is a special form of transferability estimation.
*   **vs. Traditional Transferability Estimation (TE):** Standard TE measures model-data alignment using ground-truth labels. C-MoNA measures *model-model compatibility* (teacher-student alignment) without labels.
*   **Theoretical Basis:** We provided a rigorous derivation (addressing Reviewer pwtL) showing why supervised rankings could fundamentally differ from distillation rankings. Distillation involves teacher-dependent interaction terms and capacity gaps that cause **ranking reversals**, rendering standard TE metrics ineffective.

**2. Methodological Innovation: Cross-Model NTK via JL Projections**

We addressed concerns regarding our metric's design and theoretical assumptions (Reviewers JQxR, pwtL):
*   **Cross-Model NTK:** We adapt NTK theory to measure gradient-space alignment between teacher and student. While Reviewer JQxR noted that modern training exceeds the "lazy regime," we clarified that we use NTK at *initialization* as a probe for **architectural inductive bias compatibility**, not as a full training dynamics simulator. Our strong results on deep Transformers (ViT, Swin) validate this approach.
*   **Johnson-Lindenstrauss (JL) Projections:** We justified JL projections over PCA/Autoencoders (Reviewer pwtL) because they provide theoretical distortion guarantees for preserving inner products across mismatched dimensions in a training-free, architecture-agnostic manner.

**3. State-of-the-Art Performance and Benchmarking**

We significantly expanded our evaluation to prove superiority over existing methods, including recent works like WACV 2025’s perturbation module (Reviewer pwtL).
*   **Ranking Correlation:** C-MoNA achieves **1.77$\times$ to 7.5$\times$ higher Kendall’s $\tau_w$** compared to TE baselines (SFDA, NLEEP, PACTran) across CIFAR100, SUN397, and Caltech101.
*   **Stability:** C-MoNA exhibits **3--5$\times$ lower variance** across diverse teacher architectures compared to the best baselines.
*   **Low-Data Robustness:** As requested by Reviewer KYw6, sensitivity analysis confirms C-MoNA is robust down to **50 samples (<1% data)**, where baselines often collapse or yield negative correlations.

**4. Practical Utility: Near-Oracle Selection**

To address concerns about the practical value of the metric (Reviewers LSza, pwtL), we added comprehensive top-1 and top-5 analysis:
*   **Post-Distillation Performance:** The student model selected by C-MoNA achieves downstream accuracy and stability **near-identical to the Oracle** (best possible choice). For example, on Caltech101, C-MoNA yields $92.34 \pm 0.48\%$ accuracy vs. Oracle $92.78 \pm 0.49\%$, while the next best baseline lags at $90.45 \pm 1.58\%$.
*   **Top-K Hits:** C-MoNA consistently places at least **3 out of the top 5** best students in its top-5 predictions across all datasets and teachers.

**5. Robustness to Domain Shift**

Addressing Reviewer KYw6, we demonstrated performance on Object Detection and long-tailed datasets (Caltech101). While extreme low-diversity shifts (e.g., 4-class African Wildlife) remain challenging for all metrics, C-MoNA generally maintains robust positive correlations where others fail.

**Conclusion**

This work contributes both the **formalization** of a critical practical problem—selecting compatible students for KD without labels or training—and the **first effective solution**. By bridging NTK theory with practical efficient projection methods, C-MoNA allows practitioners to bypass expensive trial-and-error fine-tuning, offering a reliable, training-free selection mechanism that outperforms all adapted baselines.

---

### Meta-Review · Area_Chair_4seS · 2026-01-07

**Summary:**

The paper proposes a principled approach for choosing the best student in the knowledge distillation setup. The reviewers appreciated the originality of the problem and the method, but raised concerns about lack of downstream results and strong baselines.

**Reviewer Concerns:**

See below.

**Reviewer Scores:**

Reviewer pwtL raised concerns around clarity, ablation studies, lack of SOTA baselines and weak practical results. I think the authors’ rebuttal sufficiently answered most of the concerns and the score would either be retained at 4 or would be increased to 6.

Reviewer JQxR raises the following concerns: 1) lack of theoretical guarantees, 2) lack of failure analysis. The rebuttal sufficiently answered the reviewer's questions and they indicated they were going to increase the score, likely 4->6.

Reviewer KYw6 raised concerns related to sensitivity analysis across independent runs, stability in domain shifts, and lack of downstream performance evaluation. I believe the authors’ rebuttal sufficiently addressed these concerns and the reviewer would increase the score 4->6.

Reviewer LSza raised concerns around lack of comparisons to KD + NAS related works, lack of downstream performance evaluation and weak practical results. I believe that the authors’ rebuttal didn’t convincingly respond to this feedback and this reviewer would retain score 4.

I share concerns of reviewer LSza and believe that this paper will benefit from a revision which will address these concerns in depth.

---

### Decision · Program_Chairs · 2026-01-26

Reject